# Equivariant Diffusion Policy

**Dian Wang**[1*], **Stephen Hart**[2], **David Surovik**[2], **Tarik Kelestemur**[2], **Haojie Huang**[1],
**Haibo Zhao**[1], **Mark Yeatman**[2], **Jiuguang Wang**[2], **Robin Walters**[1], **Robert Platt**[1,2]

[1]Northeastern University      [2]Boston Dynamics AI Institute

https://equidiff.github.io

**Abstract:** Recent work has shown diffusion models are an effective approach to learning the multimodal distributions arising from demonstration data in behavior cloning. However, a drawback of this approach is the need to learn a denoising function, which is significantly more complex than learning an explicit policy. In this work, we propose Equivariant Diffusion Policy, a novel diffusion policy learning method that leverages domain symmetries to obtain better sample efficiency and generalization in the denoising function. We theoretically analyze the $SO(2)$ symmetry of full 6-DoF control and characterize when a diffusion model is $SO(2)$-equivariant. We furthermore evaluate the method empirically on a set of 12 simulation tasks in MimicGen, and show that it obtains a success rate that is, on average, 21.9% higher than the baseline Diffusion Policy. We also evaluate the method on a real-world system to show that effective policies can be learned with relatively few training samples, whereas the baseline Diffusion Policy cannot.

**Keywords:** Equivariance, Diffusion Model, Robotic Manipulation

## 1 Introduction

The recently proposed *Diffusion Policy* [1] formulates robotic manipulation action prediction as a diffusion model that denoises the action conditioned on the observation, thereby better capturing the multimodal action distribution of the demonstration data in Behavior Cloning (BC). Although Diffusion Policy often outperforms baselines on benchmarks [2, 3], a key drawback is that the denoising function is more complex than a standard policy function. In particular, for a single state-action pair $(s, a)$, the denoising process uses a mapping $(s, a + \varepsilon^k, k) \mapsto \varepsilon^k$ for all possible $k$ and $\varepsilon^k$, where $\varepsilon^k$ is Gaussian noise conditioned on step $k$, which is harder to train compared with an explicit BC $s \mapsto a$.

In this paper, we leverage equivariant neural models to embed task symmetry as an inductive bias in the diffusion process, making the denois-

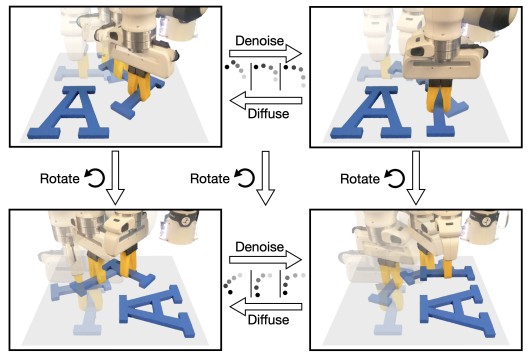

Figure 1: Equivariance in diffusion policy. Top left: a randomly sampled trajectory. Top right: a valid trajectory after denoising. If the state and the random trajectory are both rotated (bottom left), and we rotate the noise accordingly in the denoising process, we will end up with a successful trajectory in the rotated state (bottom right).

ing function easier to learn. Although equivariant diffusion models have been studied by a number of prior works [4, 5, 6, 7, 8], our paper is the first to study the idea in the context of visuomotor policy learning. As illustrated in Figure 1, rotation of a state and noisy trajectory action about the gravity axis (i.e., rotated on the tabletop) results in a corresponding rotation of the denoised trajec-

---

*Part of the work was done as an intern at the Boston Dynamics AI Institute

8th Conference on Robot Learning (CoRL 2024), Munich, Germany.

tory. As a result of this symmetry, our model is more data efficient and generalizes better than the non-symmetric baselines, mitigating the high data costs typically associated with diffusion.

Our contributions are as follows: 1) we propose Equivariant Diffusion Policy, a novel BC approach based on equivariant diffusion, 2) we analyze the conditions under which the denoising function is equivariant, 3) we theoretically demonstrate the use of $SO(2)$-equivariance in the context of 6-DoF control for robotic manipulation, which prior methods [9, 10] leveraged in a less expressive $SE(2)$ action space, and 4) we provide a thorough demonstration of our method in both simulated and physical systems. In simulation, we evaluate on 12 manipulation tasks in the MimicGen benchmark [11] and outperform the baseline Diffusion Policy by an average success rate of 21.9% when trained with 100 demos. On hardware, we show that successful policies can be learned with a small number (between 20 and 60) of demonstrations for six different manipulation tasks, including a long-horizon bagel baking task, while the original Diffusion Policy performs poorly in this low-data regime.

## 2 Related Work

**Diffusion Models**    Diffusion models [12] learn distributions by modeling the reverse of a diffusion process, which is a Markov chain that gradually adds Gaussian noise to the data until it transitions to a Gaussian distribution. Denoising diffusion models [13, 14] can be interpreted as learning the gradient field of an implicit score during training, where inference applies a sequence of score optimization steps. This new family of generative methods has proven to be effective for capturing multimodal distributions in planning [15, 16] and policy learning [17, 18, 19, 1, 20]. However, these methods did not leverage the geometric symmetries underlying the task and the diffusion process. Xu et al. [21], Hoogeboom et al. [4] show that leveraging $SO(3)$ symmetries from the domain in the diffusion process dramatically improves sample efficiency and generalization ability in molecular generation. EDGI [6] extends diffuser [15] to equivariant diffusion planning with improved performance, but relies on the ground truth state as the input. Ryu et al. [7] propose bi-equivariant diffusion models for visual robotic manipulation, while limited to open-loop settings. By contrast, we exploit domain symmetries during the diffusion process to attain an effective closed-loop visuomotor policy.

**Equivariance in manipulation policies**    Robots operate within a three-dimensional Euclidean space, where manipulation tasks inherently encompass geometric symmetries such as rotations. Recent works [9, 22, 23, 24, 25, 26, 27, 10, 28, 29, 30, 31, 32] compellingly show that improvement in sample efficiency and performance can be obtained by leveraging symmetries in policy leaning. [33, 34, 35] show the efficiency of equivariant models for on-robot learning. [36, 37, 38, 39] learn an open-loop pick and place policy with few demonstrations. While this prior work either considers symmetries in $SE(3)$ open-loop or $SE(2)$ closed-loop action spaces, our paper studies symmetries in an $SE(3)$ closed-loop action space, and is the first one to study the symmetries in diffusion policy.

**Closed-loop Visuomotor control**    Closed-loop visuomotor policies are more robust and responsive but struggle with learning from diverse trajectories and predicting long-horizon actions. Previous methods [40, 41, 42, 43] directly map from observations to actions. However, this type of explicit policy learning struggles to learn multimodal behavior distributions and may not be expressive enough to capture the full range and fidelity of trajectory data [17, 44]. Several works propose implicit policies [45, 46] with energy-based models [47, 48]. However, training is challenging due to the necessity of a substantial volume of negative samples to effectively learn an optimal energy score function for state-action pairs. Recently, [17, 1] model action generation as a conditional denoising diffusion process and demonstrate strong performance by adapting diffusion models to sequential environments. Our work builds on [1] but focuses on equivariance in the diffusion process.

## 3 Background

**Problem Statement**    We study policy learning using behavior cloning. The agent is required to learn a mapping from the observation $\mathbf{o}$ to the action $\mathbf{a}$ that mimics an expert policy. Both $\mathbf{o}$ and $\mathbf{a}$ can

contain a number of time steps, i.e., $\mathbf{o} = \{\mathbf{o}_{t-(m-1)}, \ldots, \mathbf{o}_{t-1}, \mathbf{o}_t\}, \mathbf{a} = \{\mathbf{a}_t, \mathbf{a}_{t+1}, \ldots, \mathbf{a}_{t+(n-1)}\}$ where $m$ is the number of history steps observed and $n$ is the number of future action steps. The observation contains both visual information (images or voxels) and the pose vector of the gripper.

Let $T_t \in \mathbb{R}^{4 \times 4}$ be the current SE(3) pose of the gripper in the world frame, the actions $\mathbf{a}_t$ specify a desired pose $\mathbf{A}_t \in \mathbb{R}^{4 \times 4}$ of the gripper and an open-width command $w_t \in \mathbb{R}$. The pose can be either absolute ($T_{t+1} = \mathbf{A}_t$, also called position control) or relative ($T_{t+1} = \mathbf{A}_t T_t$, also called velocity control). In order to noise and denoise via addition and subtraction as in the standard diffusion process, we vectorize the SE(3) pose $\mathbf{A}_t$ into a vector $\mathbf{a}_t$ during diffusion and denoising, and orthogonalize the noise-free action vector after denoising.

**Diffusion Policy**  Chi et al. [1] proposed Diffusion Policy to model the multimodal distribution in behavior cloning using Denoising Diffusion Probabilistic Models (DDPMs) [14]. Diffusion Policy learns a noise prediction function $\varepsilon_\theta(\mathbf{o}, \mathbf{a} + \varepsilon^k, k) = \varepsilon^k$ using a network $\varepsilon_\theta$ parameterized by $\theta$. The network is expected to predict the noise component of the input $\mathbf{a} + \varepsilon^k$. During training, transitions $(\mathbf{o}, \mathbf{a})$ are sampled from the expert dataset. Then, random noise $\varepsilon^k$ (conditioned on a randomly sampled denoising step $k$) is added to $\mathbf{a}$. The loss is $\mathcal{L} = ||\varepsilon_\theta(\mathbf{o}, \mathbf{a} + \varepsilon^k, k) - \varepsilon^k||^2$. During inference, given an observation $\mathbf{o}$, DDPM performs a sequence of $K$ denoising steps starting from a random action $\mathbf{a}^k \sim \mathcal{N}(0, 1)$ to generate an action $\mathbf{a}^0$ defined inductively by

$$\mathbf{a}^{k-1} = \alpha(\mathbf{a}^k - \gamma\varepsilon_\theta(\mathbf{o}, \mathbf{a}^k, k) + \epsilon), \tag{1}$$

where $\epsilon \sim \mathcal{N}(0, \sigma^2 I)$. $\alpha, \gamma, \sigma$ are functions of the denoising step $k$ (also known as the noise schedule). The action $\mathbf{a}^0$ is expected to be a sample from the expert policy $\pi : \mathbf{o} \mapsto \mathbf{a}$.

**Equivariance**  A function $f$ is equivariant if it commutes with the transformations of a symmetry group $G$. Specifically, $\forall g \in G$, $f(\rho_x(g)x) = \rho_y(g)f(x)$, where $\rho: G \to GL(n)$ is called the group representation that maps each group element to an $n \times n$ invertible matrix that acts on the input and output through matrix multiplication. We sometimes leave the actions implicit and write $f(gx) = gf(x)$. We mainly focus on the group SO(2) of planar rotations (i.e., rotation around the $z$-axis of the world) and its subgroup $C_u$ containing $u$ discrete rotations. There are three particular representations of SO(2) or $C_u$ that are of interest in this paper:

1) the trivial representation $\rho_0$ defines SO(2) or $C_u$ acting on an invariant scalar $x \in \mathbb{R}$ by $\rho_0(g)x = x$. 2) the irreducible representation $\rho_\omega$ defines SO(2) or $C_u$ acting on a vector $v \in \mathbb{R}^2$ by a $2 \times 2$ rotation matrix with frequency $\omega$, $\rho_\omega(g)v = \left(\begin{smallmatrix} \cos \omega g & -\sin \omega g \\ \sin \omega g & \cos \omega g \end{smallmatrix}\right)v$. 3) the regular representation $\rho_{\text{reg}}$ that defines $C_u$ acting on a vector $x \in \mathbb{R}^u$ by $u \times u$ permutation matrices. Let $g = r^m \in C_u = \{1, r^1, \ldots, r^{u-1}\}$ and $(x_1, \ldots, x_u) \in \mathbb{R}^u$. Then $\rho_{reg}(g)x = (x_{u-m+1}, \ldots, x_u, x_1, x_2, \ldots, x_{u-m})$ cyclically permutes the coordinates of $\mathbb{R}^u$.

A representation $\rho$ can also be a combination of different representations, i.e., $\rho = \rho_0^{n_0} \oplus \rho_1^{n_1} \oplus \rho_2^{n_2} \in GL(n_0 + 2n_1 + 2n_2)$. In such a case, $\rho(g)$ is an $(n_0 + 2n_1 + 2n_2) \times (n_0 + 2n_1 + 2n_2)$ block diagonal matrix that acts on $x \in \mathbb{R}^{n_0 + 2n_1 + 2n_2}$.

# 4 Method

## 4.1 Theory of Equivariant Diffusion Policy

The main contribution of this paper is a method that incorporates equivariance in the diffusion process for policy learning. As theoretical justification, we first analyze the noise prediction function and show that it is equivariant any time the expert policy that is being modeled is equivariant. This implies equivariant neural networks have the correct inductive bias to model this function.

Let $\pi : \mathbf{o} \mapsto \mathbf{a}$ be the expert policy function, and let $\varepsilon : (\mathbf{o}, \mathbf{a}^k, k) \mapsto \varepsilon^k$ be the ground truth noise prediction function associated with the expert policy such that $\varepsilon^k = \varepsilon(\mathbf{o}, \pi(\mathbf{o}) + \varepsilon^k, k)$. Assume $g \in \text{SO}(2)$ acts upon the noise $\varepsilon^k$ in the same way as it acts upon the action $\mathbf{a}$.

**Proposition 1.** *The noise prediction function $\varepsilon$ is equivariant, i.e., $\varepsilon(g\mathbf{o}, g\mathbf{a}^k, k) = g\varepsilon(\mathbf{o}, \mathbf{a}^k, k), g \in$ SO(2), when the expert policy function is SO(2)-equivariant, i.e., $\pi(g\mathbf{o}) = g\pi(\mathbf{o}), g \in$ SO(2).*

See Appendix A for the proof. Figure 2 illustrates the equivariance property of $\varepsilon$. If we infer $\varepsilon$ for all actions in the action space, we effectively acquire a gradient field towards the expert trajectory. The figure shows that such a gradient field is equivariant when the expert policy is equivariant, thus the function $\varepsilon$ is also equivariant. Notice that the figure shows the average of all action time steps.

## 4.2 SO(2) Representation on 6DoF Action

A key step in defining an Equivariant Diffusion Policy is to define how actions $\mathbf{a}_t$ transform under rotation. We describe this transformation in terms of irreducible SO(2) representations, which allows us to build the equivariance constraint into the denoising network.

**Proposition 2.** *There exist irreducible representations that describe how* SO(2) *acts on an* SE(3) *gripper action* $\mathbf{a}_t$. *In absolute pose control, let* $\mathbf{a}_t = \mathrm{Vec}_c(\mathbf{A}_t)$ *where* $\mathrm{Vec}_c$ *flattens an* SE(3) *pose* $\mathbf{A}_t \in \mathbb{R}^{4\times 4}$ *into a vector by column,* $g\mathbf{a}_t = (\rho_1 \oplus \rho_0^2)^4(g)\mathbf{a}_t$. *In relative pose control, let* $\mathbf{a}_t = \mathrm{Vec}_r(\mathbf{A}_t)$ *where* $\mathrm{Vec}_r$ *flattens* $\mathbf{A}_t$ *into a vector by row,* $g\mathbf{a}_t = P^{-1}\left[(\rho_0^6 \oplus \rho_1^4 \oplus \rho_2)(g)\right]P\mathbf{a}_t$, *where* $P$ *is a fixed change-of-basis matrix.*

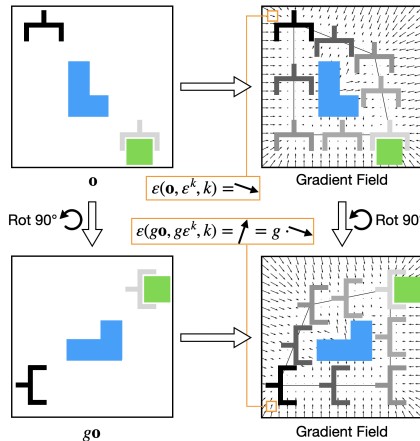

Figure 2: Equivariance of the denoising function $\varepsilon$. Left: In observation $\mathbf{o}$, the goal for the gripper is to reach the green block while avoiding the blue obstacle. Right: The expert trajectory and the gradient field associated with the denoising function. If the policy is equivariant, both the denoising function and the entire gradient field is equivariant. The orange boxes show the equivariance of $\varepsilon$ with a particular input $\varepsilon^k$.

**Absolute Control** We first consider absolute pose control, i.e., $T_{t+1} = \mathbf{A}_t$. Let $T_g$ be the transformation matrix corresponding to the SO(2) rotation along the $z$-axis of the world frame, $T_g = \begin{pmatrix} \cos g & -\sin g & 0 & 0 \\ \sin g & \cos g & 0 & 0 \\ 0 & 0 & 1 & 0 \\ 0 & 0 & 0 & 1 \end{pmatrix} = \begin{pmatrix} \rho_1(g) & & \\ & \rho_0(g) & \\ & & \rho_0(g) \end{pmatrix}$, where $\rho_1(g) = \begin{pmatrix} \cos g & -\sin g \\ \sin g & \cos g \end{pmatrix}$. The SO(2) action on $\mathbf{A}_t$ is $g\mathbf{A}_t = T_g\mathbf{A}_t = (\rho_1 \oplus \rho_0^2)(g)\mathbf{A}_t$. Vectorizing $\mathbf{A}_t$ by column gives $\mathbf{a}_t = \mathrm{Vec}_c(\mathbf{A}_t) = \left[\mathbf{A}_t^{1T}, \mathbf{A}_t^{2T}, \mathbf{A}_t^{3T}, \mathbf{A}_t^{4T}\right]^T$ where $\mathbf{A}_t^i$ is the $i$th column of $\mathbf{A}_t$. By the rule of matrix multiplication, we have $g\mathbf{A}_t^i = (\rho_1 \oplus \rho_0^2)(g)\mathbf{A}_t^i$ and $g\mathbf{a}_t = (\rho_1 \oplus \rho_0^2)^4(g)\mathbf{a}_t$.

Since the gripper open width is invariant, $gw_t = \rho_0(g)w_t$, we can append $w_t$ to $\mathbf{a}_t$ and add an extra $\rho_0$ to the representation. We can also simplify the representation by removing the constants in the transformation matrix and removing the last row in the rotation part of the transformation matrix (i.e., the 6D rotation representation [49]). The resulting action vector would be $\mathbf{a}_t \in \mathbb{R}^6 \times \mathbb{R}^3 \times \mathbb{R}$, where the first six elements are the 6D rotation, the following three elements are the translation, and the last element is the gripper open width. In such a case, we have $g\mathbf{a}_t = (\rho_1^3 \oplus (\rho_1 \oplus \rho_0) \oplus \rho_0)(g)\mathbf{a}_t$.

**Relative Control** For relative gripper pose, i.e., $T_{t+1} = \mathbf{A}_t T_t$, the group action on $\mathbf{A}_t$ satisfies $(g\mathbf{A}_t)T_g T_t = T_g(\mathbf{A}_t T_t)$ (because the rotation $g \in$ SO(2) applies to both the current pose and the change of pose). Solving for $g\mathbf{A}_t$ we get $g\mathbf{A}_t = T_g\mathbf{A}_t T_g^{-1}$. Let $\mathbf{a}_t = \mathrm{Vec}_r(\mathbf{A}_t)$ where $\mathrm{Vec}_r : \mathbb{R}^{n\times m} \to \mathbb{R}^{(n\cdot m)}$ flattens a matrix into a vector by row. Here we want to find a linear action $\rho_{\mathbf{A}}$ that satisfies $g\mathbf{a}_t = \rho_{\mathbf{A}}(g)\mathrm{Vec}_r(\mathbf{A}_t) = \mathrm{Vec}_r(T_g\mathbf{A}_t T_g^{-1})$. After solving for $\rho_{\mathbf{A}} \in \mathbb{R}^{16\times 16}$ and calculating a change-of-basis matrix $P$ such that $P\rho_{\mathbf{A}}P^{-1}$ is a block diagonal matrix consisting of irreducible representations, we have $g\mathbf{a}_t = P^{-1}\left[(\rho_0^6 \oplus \rho_1^4 \oplus \rho_2)(g)\right]P\mathbf{a}_t$ (see Appendix B for details). For easier implementation, we add a $\rho_0$ for the gripper action $w_t$, remove the constants in the transformation matrix, and decompose SE(3) = SO(3) $\times \mathbb{R}^3$. See Appendix C.

## 4.3 Implementation of Equivariant Diffusion Policy

Now that we have the theoretical grounding of the equivariance in the noise prediction function $\varepsilon$, this section will introduce the network architecture of our Equivariant Diffusion Policy. As is

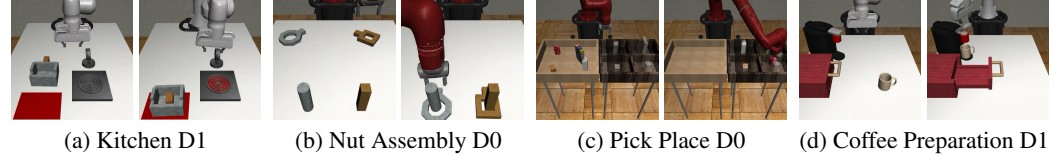

(a) Kitchen D1     (b) Nut Assembly D0     (c) Pick Place D0     (d) Coffee Preparation D1

Figure 4: The experimental environments from MimicGen [11]. The left image in each subfigure shows the initial state of the environment; the right image shows the goal state. See Figure 8 in the Appendix for all environments.

shown in Figure 3, our network consists of three main parts: encoding (white box), denoising (yellow box), and decoding (gray box). We implement our network using the escnn library [50]. First, an equivariant observation encoder and an equivariant action encoder take inputs $\mathbf{o}$ and $\mathbf{a}^k$, respectively, to create equivariant embeddings $e_{\mathbf{o}}$ and $e_{\mathbf{a}^k}$. The embeddings will be in the form of a regular representation of the subgroup $C_u \subset \mathrm{SO}(2)$ (where $u$ is the number of discrete rotations in the group). The embeddings have shape $e_{\mathbf{o}} \in \mathbb{R}^{u \times d_o}$ and $e_{\mathbf{a}^k} \in \mathbb{R}^{u \times d_a}$, where each of the $d_o$ or $d_a$ dimensional vectors encodes the features for a specific group element (i.e., a rotation angle). Second, in the denoising step, let $e_{\mathbf{o}}^g \in \mathbb{R}^{d_o}$ and $e_{\mathbf{a}^k}^g \in \mathbb{R}^{d_a}$ be a pair of partial embeddings corresponding to the same group element $g$. We process each pair with a 1D Temporal U-Net (adopted from the prior works [15, 1]) to calculate an equivariant noise embedding. Specifically, letting $k$ be the denoising step, $U$ the U-Net, and $z$ its output, we have $z^g = U(e_{\mathbf{o}}^g, e_{\mathbf{a}^k}^g, k)$. Since the same

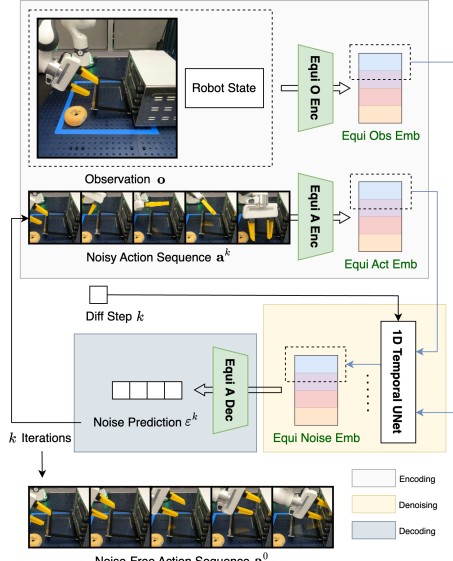

Figure 3: Overview of our Equivariant Diffusion Policy architecture.

network is applied for all $g \in C_u$, the output is an equivariant embedding of the noise in the regular representation. Finally, an equivariant decoder will decode the noise $\varepsilon^k$. See Appendix D for details.

## 5 Experiments

### 5.1 Simulation Experiment

**Experimental Settings** We first evaluate our Equivariant Diffusion Policy (EquiDiff) with either image (Im) or voxel (Vo) input on 12 manipulation tasks from MimicGen [11] (Figure 4). We define the rotation of the observation as a voxel grid rotation or an image rotation. Notice that in the image version of our method, there is a mismatch between the rotation of the agent view image and the rotation of the ground truth state since the agent view is not orthogonally top-down. Although top-down observations could be captured, we use the observation settings in the published dataset from MimicGen [11] to demonstrate the generalizability of our method[1]. On the other hand, the voxel version eliminates this symmetry mismatch as the rotation of the voxel grid aligns with the rotation of the ground truth state. To better leverage the equivariance, we also add a rotation augmentation in the voxel version of our method following our analysis in Section 4.1-4.2. We compare our method with the following baselines: 1) DiffPo-C: the original diffusion policy [1] trained with the 1D Temporal UNet [15]. Notice that the baseline shares the same UNet architecture as our method, but it does not have any equivariant structure. 2) DiffPo-T: same as above, but trained with a transformer. 3) DP3: the 3D diffusion policy [20] trained with a point net encoder. 4) ACT: the Action Chunking

---

[1]Prior work [52] demonstrate that the equivariant CNN is still able to capture symmetry in such a scenario.

Table 1, section 1:

| Method | Ctrl | Obs | Stack D1 | | | Stack Three D1 | | | Square D2 | | | Threading D2 | | |
|---|---|---|---|---|---|---|---|---|---|---|---|---|---|---|
| | | | 100 | 200 | 1000 | 100 | 200 | 1000 | 100 | 200 | 1000 | 100 | 200 | 1000 |
| EquiDiff (Vo) | Abs | Voxel | 99 (+23) | 100 (+3) | 100 (=) | 75 (+37) | 91 (+19) | 91 (-3) | 39 (+31) | 48 (+29) | 63 (+14) | 39 (+22) | 53 (+18) | 55 (-4) |
| EquiDiff (Im) | | RGB | 93 (+17) | 100 (+3) | 100 (=) | 55 (+17) | 77 (+5) | 96 (+2) | 25 (+17) | 41 (+22) | 60 (+11) | 22 (+5) | 40 (+5) | 59 (=) |
| DiffPo-C [1] | | RGB | 76 | 97 | 100 | 38 | 72 | 94 | 8 | 19 | 46 | 17 | 35 | 59 |
| DiffPo-T [1] | | RGB | 51 | 83 | 99 | 17 | 41 | 84 | 5 | 11 | 45 | 11 | 18 | 41 |
| DP3 [20] | | PCD | 69 | 87 | 99 | 7 | 23 | 65 | 7 | 6 | 19 | 12 | 23 | 40 |
| ACT [51] | | RGB | 35 | 73 | 96 | 6 | 37 | 78 | 6 | 18 | 49 | 10 | 21 | 35 |
| EquiDiff (Vo) | Rel | Voxel | 95 (+14) | 100 (+5) | 100 (=) | 59 (+33) | 76 (+24) | 83 (-9) | 25 (+17) | 35 (+14) | 52 (-7) | 33 (+20) | 39 (+13) | 46 (-1) |
| EquiDiff (Im) | | RGB | 75 (-6) | 96 (+1) | 100 (=) | 25 (-1) | 63 (+11) | 92 (=) | 11 (+3) | 21 (=) | 48 (-11) | 11 (-2) | 22 (-4) | 49 (+2) |
| DiffPo-C [1] | | RGB | 81 | 93 | 99 | 26 | 52 | 86 | 6 | 13 | 37 | 13 | 26 | 40 |
| BC RNN [2] | | RGB | 59 | 95 | 100 | 12 | 48 | 92 | 8 | 21 | 59 | 7 | 13 | 47 |

Table 1, section 2:

| Method | Ctrl | Obs | Coffee D2 | | | Three Pc. Assembly D2 | | | Hammer Cleanup D1 | | | Mug Cleanup D1 | | |
|---|---|---|---|---|---|---|---|---|---|---|---|---|---|---|
| | | | 100 | 200 | 1000 | 100 | 200 | 1000 | 100 | 200 | 1000 | 100 | 200 | 1000 |
| EquiDiff (Vo) | Abs | Voxel | 65 (+18) | 73 (+7) | 76 (-3) | 37 (+33) | 58 (+52) | 71 (+28) | 70 (+16) | 66 (-5) | 73 (-14) | 53 (+10) | 65 (+6) | 68 (+5) |
| EquiDiff (Im) | | RGB | 60 (+13) | 79 (+13) | 76 (-3) | 15 (+11) | 39 (+33) | 69 (+26) | 65 (+11) | 63 (-8) | 77 (-10) | 49 (+6) | 64 (+5) | 67 (+2) |
| DiffPo-C [1] | | RGB | 44 | 66 | 79 | 4 | 6 | 30 | 52 | 59 | 73 | 43 | 59 | 65 |
| DiffPo-T [1] | | RGB | 47 | 61 | 75 | 1 | 4 | 43 | 48 | 60 | 76 | 30 | 43 | 63 |
| DP3 [20] | | PCD | 34 | 45 | 69 | 0 | 1 | 3 | 54 | 71 | 87 | 21 | 33 | 53 |
| ACT [51] | | RGB | 19 | 33 | 64 | 0 | 3 | 24 | 38 | 54 | 71 | 23 | 31 | 56 |
| EquiDiff (Vo) | Rel | Voxel | 55 (+12) | 59 (+7) | 64 (-12) | 5 (+3) | 5 (=) | 55 (+28) | 64 (+21) | 62 (+8) | 67 (-5) | 39 (+14) | 43 (+4) | 62 (-5) |
| EquiDiff (Im) | | RGB | 41 (-2) | 59 (+7) | 66 (-10) | 1 (-1) | 5 (=) | 59 (+32) | 49 (+6) | 52 (-2) | 69 (-3) | 29 (+4) | 36 (-3) | 65 (-2) |
| DiffPo-C [1] | | RGB | 43 | 51 | 67 | 2 | 2 | 20 | 43 | 54 | 65 | 25 | 39 | 55 |
| BC RNN [2] | | RGB | 37 | 52 | 76 | 0 | 5 | 27 | 32 | 43 | 72 | 19 | 39 | 67 |

Table 1, section 3:

| Method | Ctrl | Obs | Kitchen D1 | | | Nut Assembly D0 | | | Pick Place D0 | | | Coffee Preparation D1 | | |
|---|---|---|---|---|---|---|---|---|---|---|---|---|---|---|
| | | | 100 | 200 | 1000 | 100 | 200 | 1000 | 100 | 200 | 1000 | 100 | 200 | 1000 |
| EquiDiff (Vo) | Abs | Voxel | 85 (+18) | 89 (+4) | 88 (-3) | 67 (+12) | 77 (+9) | 83 (-1) | 58 (+23) | 68 (+3) | 82 (-1) | 80 (+15) | 83 (+21) | 85 (+9) |
| EquiDiff (Im) | | RGB | 67 (=) | 77 (-8) | 81 (-10) | 74 (+19) | 85 (+17) | 94 (+10) | 42 (+7) | 74 (+9) | 92 (+9) | 77 (+12) | 83 (+21) | 85 (+9) |
| DiffPo-C [1] | | RGB | 67 | 85 | 87 | 55 | 68 | 83 | 35 | 65 | 83 | 65 | 62 | 58 |
| DiffPo-T [1] | | RGB | 54 | 75 | 81 | 31 | 32 | 46 | 15 | 37 | 50 | 38 | 51 | 76 |
| DP3 [20] | | PCD | 45 | 71 | 91 | 16 | 24 | 58 | 12 | 15 | 34 | 10 | 22 | 63 |
| ACT [51] | | RGB | 37 | 61 | 87 | 42 | 64 | 84 | 7 | 17 | 50 | 32 | 46 | 65 |
| EquiDiff (Vo) | Rel | Voxel | 69 (+27) | 83 (+19) | 89 (+8) | 53 (+11) | 65 (+3) | 72 (-13) | 40 (+5) | 58 (-1) | 79 (-3) | 48 (+6) | 71 (+18) | 73 (+12) |
| EquiDiff (Im) | | RGB | 61 (+19) | 72 (+8) | 83 (+2) | 44 (+2) | 65 (+3) | 87 (+2) | 29 (-6) | 55 (-4) | 91 (+9) | 49 (+7) | 59 (+6) | 79 (+18) |
| DiffPo-C [1] | | RGB | 42 | 64 | 81 | 42 | 62 | 75 | 35 | 59 | 82 | 42 | 53 | 51 |
| BC RNN [2] | | RGB | 31 | 47 | 81 | 35 | 58 | 85 | 21 | 41 | 77 | 14 | 32 | 61 |

Table 1: The performance of our method compared with the baselines in simulation. We experiment with 100, 200, and 1000 demos in each environment and report the maximum task success rate among 50 evaluations throughout training. Results averaged over three seeds. Number in parentheses shows the difference between our method and the best baseline (with increment colored in blue and decrement in red). Bold performance indicates the best, bold difference is greater than 10%.

Transformer [51] trained as a conditional VAE. 5) BC RNN: a recurrent architecture from [2]. Notice that the voxel version of our method and DP3 utilizes the 3D inputs constructed from four cameras, while the image version of our method and the other baselines directly use the RGB images from two cameras. As our main baseline, we evaluate DiffPo-C in both absolute and relative pose control. We evaluate the other baselines in the same control mode as in the original work (absolute for DiffPo-T, DP3 and ACT, and relative for BC RNN). See Appendix E and F for the details.

**Results**    Table 1 shows the experimental result in terms of the maximum success rate among 50 evaluations throughout the training. First, with absolute pose control, our Equivariant Diffusion Policy with voxel input achieves the best overall performance, outperforming the baselines in 11 out of the 12 environments (Hammer Cleanup D1 being the exception). With RGB image inputs, our method outperforms all RGB baselines in all environments except for Kitchen D1. Second, in relative control, our method with voxel input achieves the best performance, while our method with RGB input is only marginally better than the baselines. Third, our method appears to perform particularly well in the low-

| Method | Ctrl | Average over 12 Environments | | |
|---|---|---|---|---|
| | | 100 | 200 | 1000 |
| EquiDiff (Vo) | Abs | 63.9 (+21.9) | 72.6 (+14.8) | 77.9 (+6.5) |
| EquiDiff (Im) | | 53.7 (+11.7) | 68.5 (+10.7) | 79.7 (+8.3) |
| DiffPo-C [1] | | 42.0 | 57.8 | 71.4 |
| DiffPo-T [1] | | 29.0 | 43.0 | 64.9 |
| DP3 [20] | | 23.9 | 35.1 | 56.8 |
| ACT [51] | | 21.3 | 38.2 | 63.3 |
| EquiDiff (Vo) | Rel | 48.8 (+15.5) | 58.0 (+10.7) | 70.2 (-0.1) |
| EquiDiff (Im) | | 35.4 (+2.1) | 50.4 (+3.1) | 74.0 (+3.7) |
| DiffPo-C [1] | | 33.3 | 47.3 | 63.2 |
| BC RNN [2] | | 22.9 | 41.2 | 70.3 |

Table 2: The average performance over 12 tasks of Equivariant Diffusion Policy compared with baselines.

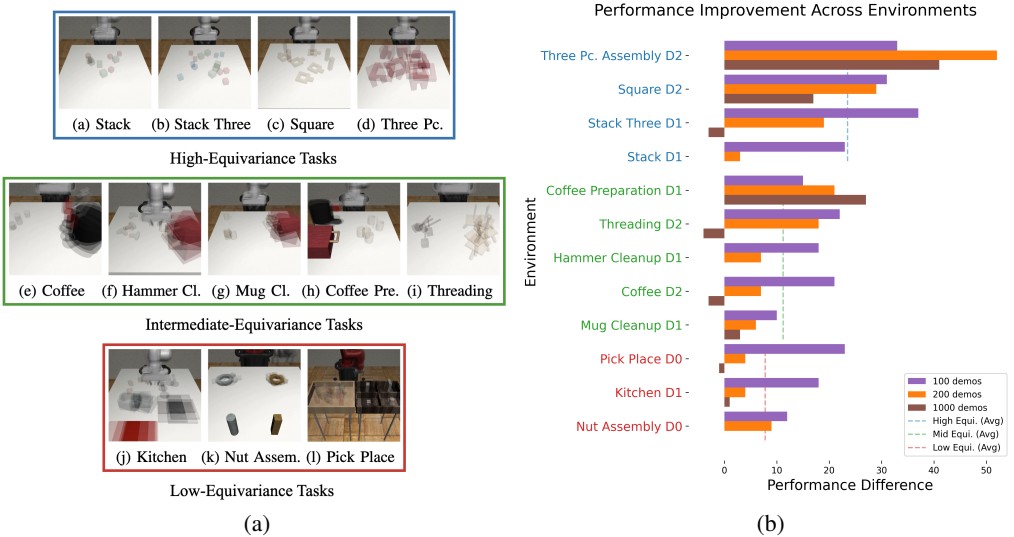

Figure 5: (a) The three task groups are based on the level of equivariance and their initial object distribution. Images were generated by taking the average of five random initialization states. (b) The performance improvement of our Equivariant Diffusion Policy (Voxel) compared with the original diffusion policy in absolute pose control. Blue environments are high-equivariance tasks; green environments are intermediate-equivariance tasks; red environments are low-equivariance tasks.

data regime (i.e., with 100 or 200 demos). Specifically, taking the average over all environments (as is shown in Table 2), our method with voxel input and absolute pose control trained with 100 demos outperforms the best baseline by 21.9%. When trained with 200 demos, it outperforms all baselines trained with 1000 demos, indicating the strong sample efficiency of our method. We also perform an ablation study in Appendix H, where we ablate the equivariant structure and the voxel input in our method. We show that though both of these items contribute to the performance improvement our method shows over the baseline, the equivariant structure is the more important factor.

## 5.2 Improvement with Different Levels of Equivariance

We further analyze the performance improvement of our method when the tasks have different levels of equivariance. Since equivariant models generalize automatically across different object poses, equivariance should hypothetically be more useful when there is greater variance in the distribution of initial object poses. We qualitatively group the tasks into three levels: 1) high-equivariance tasks where the poses of the objects are initialized randomly within the workspace; 2) intermediate-equivariance tasks where each object is initialized in a certain range, but with some randomness inside the range; 3) low-equivariance tasks where there is no randomness for the position and/or orientation of certain objects. Figure 5a shows the three task groups. We show the performance improvement of our Equivariant Diffusion Policy with voxel in absolute pose control compared with the standard diffusion policy in Figure 5b. Generally, the high-equivariance tasks benefit more from injecting symmetry in the network architecture. Moreover, our method's strong performance in the intermediate and low-equivariance tasks indicates its robustness and generalizability, as the model's symmetry is helpful even when the task is partially symmetric.

## 5.3 Real-Robot Experiment

**Experimental Settings**  In this section, we evaluate our method on a real robot system containing a Franka Emika robot arm [53] equipped with a pair of fin-ray [54] fingers and three Intel Realsense [55] D455 cameras. Demonstrations were gathered by an operator using a 6DoF 3DConnexion mouse. Observations and demonstration actions were recorded at 5Hz. Similarly to prior

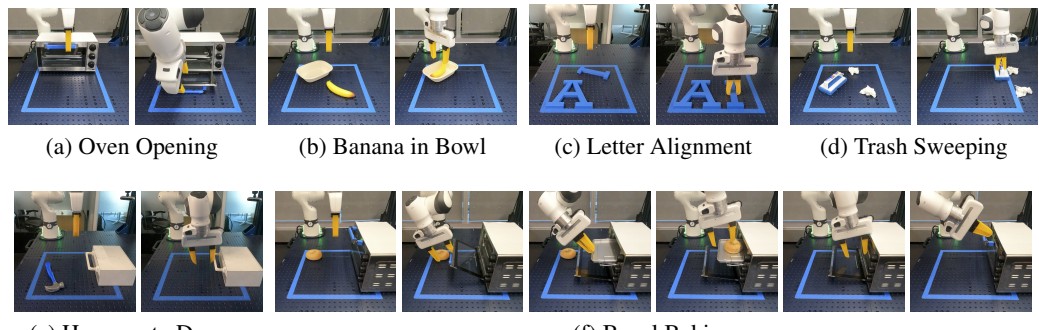

| (a) Oven Opening | (b) Banana in Bowl | (c) Letter Alignment | (d) Trash Sweeping |
| --- | --- | --- | --- |

| (e) Hammer to Drawer | (f) Bagel Baking |
| --- | --- |

Figure 6: The real-world environments. The left image of each subfigure shows the initial state of the environment; the right image shows the goal state. See Appendix L for a detailed task description.

| | Oven Opening | Banana in Bowl | Letter Alignment | Trash Sweeping | Hammer to Drawer | Bagel Baking |
| --- | --- | --- | --- | --- | --- | --- |
| # Demos | 20 | 40 | 40 | 40 | 60 | 58 |
| EquiDiff (Vo) | 95% (19/20) | 95% (19/20) | 95% (19/20) | 90% (18/20) | 85% (17/20) | 80% (16/20) |
| DiffPo-C (Vo) | 60% (12/20) | 30% (6/20) | 0% (0/20) | 5%(1/20) | 5%(1/20) | 10% (2/20) |

Table 3: Performance of Equivariant Diffusion Policy in Real-World Robot Experiments.

work [1], we use DDIM [56] in this experiment to reduce the number of denoising steps to 16. Figure 6 shows the six tasks in this experiment. We compare our Equivariant Diffusion Policy with voxel input against a baseline Diffusion Policy, which uses the same voxel grid as the vision input and uses a non-equivariant 3D convolutional encoder with approximately the same number of trainable parameters as ours. As we show in the ablation study (Appendix H), this baseline works better than the original diffusion policy with image input.

**Results**  We evaluate the trained models over 20 test trials for each task. The results are shown in Table 3. Our Equivariant Diffusion Policy can solve those tasks with only 20 to 60 demonstrations. Notably, our method achieves an 80% success rate in bagel baking, where the failures were all due to the joint limits of the robot. In comparison, the baseline performs poorly in all six tasks.

## 6 Conclusion

This paper studies the leveraging of symmetries in visuomotor policy learning. We propose the novel Equivariant Diffusion Policy method and provide a theoretical analysis identifying the conditions under which diffusion processes are equivariant. We also demonstrate a general framework for using $SO(2)$-equivariance in the 6DoF control for robotic manipulation. We evaluate our method in both simulation and the real world and show in both cases that our method outperforms the baseline Diffusion Policy by a large margin.

One limitation of this work is the partial utilization of the power of equivariance due to the symmetry mismatch in the vision system. Even with the voxel input, Factors like the arm's occasional presence in the voxel grid and camera noise could break symmetry. Future work could address this by designing a vision system free of symmetry corruption. Additionally, "incorrect equivariance", as shown in prior work [57], may harm performance when the model's symmetry conflicts with the demonstration. Another limitation is that although the theory in Section 4.2 is not limited to diffusion policies and can apply to other policy learning pipelines as well, this is not demonstrated. Specifically, given the good performance of BC RNN with the relative pose control in Table 1, experimenting with an equivariant version of BC RNN could be beneficial. Finally, extending our method to other robotic tasks like navigation, locomotion, and mobile manipulation is a key future direction.

**Acknowledgments**

This work is supported in part by NSF 1750649, NSF 2107256, NSF 2314182, NSF 2134178, NSF 2409351, and NASA 80NSSC19K1474. Dian Wang is supported in part by the JPMorgan Chase PhD fellowship. The authors would like to thank Dr. Osman Dogan Yirmibesoglu for the design of the fin-ray gripper fingers, Dr. Andy Park for building the teleop system for data collection, Emmanuel Panov for collecting demonstration data in the robot experiment, Dr. Thomas Weng for the proof-read of the paper, and Dr. Cheng Chi for the helpful discussion.

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

# A Proof of Proposition 1

*Proof.* Consider the observation $\mathbf{o}$ and the action $\mathbf{a} = \pi(\mathbf{o})$, let $\mathbf{a}^k = \mathbf{a} + \varepsilon^k$. By the definition of the noise prediction function $\varepsilon$, we have

$$\varepsilon^k = \varepsilon(\mathbf{o}, \pi(\mathbf{o}) + \varepsilon^k, k). \tag{2}$$

Applying $g \in \mathrm{SO}(2)$ to $\mathbf{o}$ we have

$$\varepsilon^k = \varepsilon(g\mathbf{o}, \pi(g\mathbf{o}) + \varepsilon^k, k). \tag{3}$$

Since $\pi$ is equivariant,

$$\varepsilon^k = \varepsilon(g\mathbf{o}, g\mathbf{a} + \varepsilon^k, k). \tag{4}$$

Since the noise prediction function predicts the noise as long as $\varepsilon^k$ is the same on both sides of the equation, we can substitute $\varepsilon^k$ with $g\varepsilon^k$

$$g\varepsilon^k = \varepsilon(g\mathbf{o}, g\mathbf{a} + g\varepsilon^k, k). \tag{5}$$

By linearity, $g\mathbf{a} + g\varepsilon^k = g(\mathbf{a} + \varepsilon^k) = g\mathbf{a}^k$ and thus

$$g\varepsilon^k = \varepsilon(g\mathbf{o}, g\mathbf{a}^k, k). \tag{6}$$

Replacing $\varepsilon^k$ with $\varepsilon(\mathbf{o}, \mathbf{a}^k, k)$ gives $g\varepsilon(\mathbf{o}, \mathbf{a}^k, k) = \varepsilon(g\mathbf{o}, g\mathbf{a}^k, k)$ as desired. $\qquad\square$

# B Decomposing Group Representation in Relative Pose Control into Irreducible Representations

In Section 4.2, we want to find a linear action $\rho_{\mathbf{A}}$ that satisfies $g\mathbf{a}_t = \rho_{\mathbf{A}}(g)\mathrm{Vec}_r(\mathbf{A}_t) = \mathrm{Vec}_r(T_g \mathbf{A}_t T_g^{-1})$. solving for $\rho_{\mathbf{A}} \in \mathbb{R}^{16 \times 16}$ we have the group action of $\mathrm{SO}(2)$ on $\mathrm{Vec}_r(\mathbf{A}_t)$ as

$$\rho_{\mathbf{A}} = \begin{bmatrix}
c^2 & -\frac{s_2}{2} & 0 & 0 & -\frac{s_2}{2} & s^2 & 0 & 0 & & & & \\
\frac{s_2}{2} & c^2 & 0 & 0 & -s^2 & -\frac{s_2}{2} & 0 & 0 & & & & \\
0 & 0 & c & 0 & 0 & 0 & -s & 0 & & & & \\
0 & 0 & 0 & c & 0 & 0 & 0 & -s & & & & \\
\frac{s_2}{2} & -s^2 & 0 & 0 & c^2 & -\frac{s_2}{2} & 0 & 0 & & & & \\
s^2 & \frac{s_2}{2} & 0 & 0 & \frac{s_2}{2} & c^2 & 0 & 0 & & & & \\
0 & 0 & s & 0 & 0 & 0 & c & 0 & & & & \\
0 & 0 & 0 & s & 0 & 0 & 0 & c & & & & \\
& & & & & & & & \rho_1(g) & & & \\
& & & & & & & & & I_2 & & \\
& & & & & & & & & & \rho_1(g) & \\
& & & & & & & & & & & I_2
\end{bmatrix}, \tag{7}$$

where $c = \cos g, s = \sin g, c_2 = \cos 2g, s_2 = \sin 2g$. Define $P$ as

$$P = \begin{bmatrix}
1 & 0 & 0 & 0 & 0 & 1 & 0 & 0 & 0 & 0 & 0 & 0 & 0 & 0 & 0 & 0 \\
0 & 1 & 0 & 0 & -1 & 0 & 0 & 0 & 0 & 0 & 0 & 0 & 0 & 0 & 0 & 0 \\
0 & 0 & 0 & 0 & 0 & 0 & 0 & 0 & 0 & 0 & 1 & 0 & 0 & 0 & 0 & 0 \\
0 & 0 & 0 & 0 & 0 & 0 & 0 & 0 & 0 & 0 & 0 & 1 & 0 & 0 & 0 & 0 \\
0 & 0 & 0 & 0 & 0 & 0 & 0 & 0 & 0 & 0 & 0 & 0 & 0 & 0 & 1 & 0 \\
0 & 0 & 0 & 0 & 0 & 0 & 0 & 0 & 0 & 0 & 0 & 0 & 0 & 0 & 0 & 1 \\
0 & 0 & 1 & 0 & 0 & 0 & 0 & 0 & 0 & 0 & 0 & 0 & 0 & 0 & 0 & 0 \\
0 & 0 & 0 & 0 & 0 & 0 & 1 & 0 & 0 & 0 & 0 & 0 & 0 & 0 & 0 & 0 \\
0 & 0 & 0 & 1 & 0 & 0 & 0 & 0 & 0 & 0 & 0 & 0 & 0 & 0 & 0 & 0 \\
0 & 0 & 0 & 0 & 0 & 0 & 0 & 1 & 0 & 0 & 0 & 0 & 0 & 0 & 0 & 0 \\
0 & 0 & 0 & 0 & 0 & 0 & 0 & 0 & 1 & 0 & 0 & 0 & 0 & 0 & 0 & 0 \\
0 & 0 & 0 & 0 & 0 & 0 & 0 & 0 & 0 & 1 & 0 & 0 & 0 & 0 & 0 & 0 \\
0 & 0 & 0 & 0 & 0 & 0 & 0 & 0 & 0 & 0 & 0 & 0 & 1 & 0 & 0 & 0 \\
0 & 0 & 0 & 0 & 0 & 0 & 0 & 0 & 0 & 0 & 0 & 0 & 0 & 1 & 0 & 0 \\
0 & 1 & 0 & 0 & 1 & 0 & 0 & 0 & 0 & 0 & 0 & 0 & 0 & 0 & 0 & 0 \\
-1 & 0 & 0 & 0 & 0 & 1 & 0 & 0 & 0 & 0 & 0 & 0 & 0 & 0 & 0 & 0
\end{bmatrix}, \tag{8}$$

We then have

$$P\rho_{\mathbf{A}}P^{-1} = \begin{bmatrix} I_6 & & & & & \\ & \rho_1(g) & & & & \\ & & \rho_1(g) & & & \\ & & & \rho_1(g) & & \\ & & & & \rho_1(g) & \\ & & & & & \rho_2(g) \end{bmatrix}. \tag{9}$$

## C  Simplifying Group Action in Relative Pose Control

In Section 4.2, we want to find a linear action $\rho_{\mathbf{A}}$ that satisfies

$$\rho_{\mathbf{A}}(g)\text{Vec}_r(\mathbf{A}_t) = \text{Vec}_r(T_g \mathbf{A}_t T_g^{-1}), \tag{10}$$

To simplify the problem, we decompose $\mathbf{A}_t = \begin{bmatrix} \mathbf{R}_t & \mathbf{D}_t \\ 0 & 1 \end{bmatrix}$ where $\mathbf{R}_t$ is the SO(3) rotation and $\mathbf{D}_t = [x, y, z]^T$ is the translation. Define $R_g$ as the rotation matrix in $T_g$,

$$R_g = \begin{bmatrix} \cos g & -\sin g & 0 \\ \sin g & \cos g & 0 \\ 0 & 0 & 1 \end{bmatrix} = \begin{bmatrix} \rho_1(g) & \\ & \rho_0(g) \end{bmatrix}. \tag{11}$$

Since the conjugate does not apply to translation, we can write $g\mathbf{D}_t = R_g \mathbf{D}_t = [\rho_1(x, y), \rho_0(z)]^T$

For rotation, similar as before, we need to find the representation $\rho_{\mathbf{R}}$ that satisfies

$$\rho_{\mathbf{R}}(g)\text{Vec}_r(\mathbf{R}_t) = \text{Vec}_r(R_g \mathbf{R}_t R_g^{-1}). \tag{12}$$

Solving for $\rho_{\mathbf{R}}(g) \in \mathbb{R}^{9 \times 9}$ we have

$$\rho_{\mathbf{R}} = \begin{bmatrix} c^2 & -cs & 0 & -cs & s^2 & 0 & 0 & 0 & 0 \\ cs & c^2 & 0 & -s^2 & -cs & 0 & 0 & 0 & 0 \\ 0 & 0 & c & 0 & 0 & -s & 0 & 0 & 0 \\ cs & -s^2 & 0 & c^2 & -cs & 0 & 0 & 0 & 0 \\ s^2 & cs & 0 & cs & c^2 & 0 & 0 & 0 & 0 \\ 0 & 0 & s & 0 & 0 & c & 0 & 0 & 0 \\ 0 & 0 & 0 & 0 & 0 & 0 & c & -s & 0 \\ 0 & 0 & 0 & 0 & 0 & 0 & s & c & 0 \\ 0 & 0 & 0 & 0 & 0 & 0 & 0 & 0 & 1 \end{bmatrix}, \tag{13}$$

where $c = \cos g, s = \sin g$. To decompose it into the irreducible representations of SO(2), we define

$$P = \begin{bmatrix} 1 & 0 & 0 & 0 & 1 & 0 & 0 & 0 & 0 \\ 0 & -1 & 0 & 1 & 0 & 0 & 0 & 0 & 0 \\ 0 & 0 & 0 & 0 & 0 & 0 & 0 & 0 & 1 \\ 0 & 0 & 1 & 0 & 0 & 0 & 0 & 0 & 0 \\ 0 & 0 & 0 & 0 & 0 & 1 & 0 & 0 & 0 \\ 0 & 0 & 0 & 0 & 0 & 0 & 1 & 0 & 0 \\ 0 & 0 & 0 & 0 & 0 & 0 & 0 & 1 & 0 \\ 0 & 1 & 0 & 1 & 0 & 0 & 0 & 0 & 0 \\ -1 & 0 & 0 & 0 & 1 & 0 & 0 & 0 & 0 \end{bmatrix}. \tag{14}$$

Such that $P\rho_{\mathbf{R}}P^{-1}$ is a block diagonal matrix consisting of irreducible representations

$$P\rho_{\mathbf{R}}P^{-1} = \begin{bmatrix} \rho_0(g) & & & & & \\ & \rho_0(g) & & & & \\ & & \rho_0(g) & & & \\ & & & \rho_1(g) & & \\ & & & & \rho_1(g) & \\ & & & & & \rho_2(g) \end{bmatrix}. \tag{15}$$

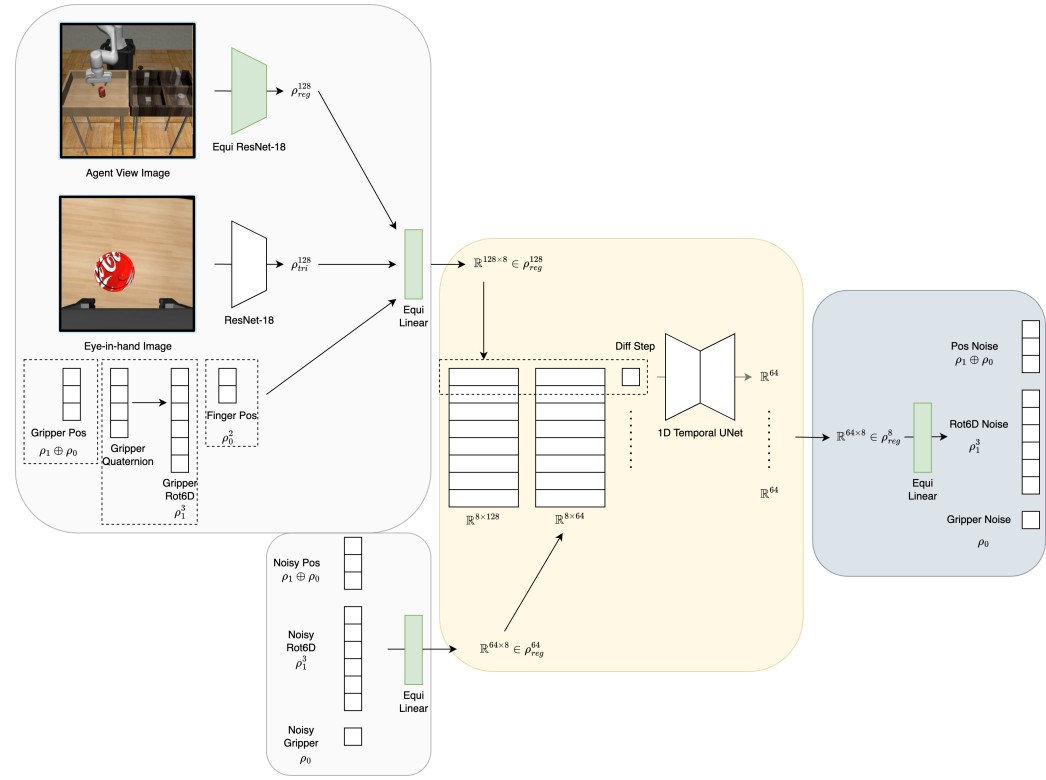

Figure 7: The detailed network architecture of our Equivariant Diffusion Policy in the simulation experiments.

We can then use $\rho(g) = P\rho_{\mathbf{R}}P^{-1} = \rho_0^3(g) \oplus \rho_1^2(g) \oplus \rho_2(g) \in \mathbb{R}^{9\times 9}$ as the group representation of the output of the equivariant network, then construct the $3 \times 3$ rotation matrix $\mathbf{R}_t$ using $P$. Specifically, let $V \in \mathbb{R}^9$ be the output of the network associated with the representation $\rho(g)$ (i.e., $g$ acts on $V$ through $\rho(g)V$). Define

$$\text{Vec}_r(\mathbf{R}_t) = P^{-1}V. \tag{16}$$

Applying $\rho(g)$ on $V$ will lead to

$$P^{-1}\rho(g)V \tag{17}$$
$$= P^{-1}P\rho_{\mathbf{R}}P^{-1}V \tag{18}$$
$$= \rho_{\mathbf{R}}\text{Vec}_r(\mathbf{R}), \tag{19}$$

which is the desired property in the equivariant network.

In the end, adding the group action for the translation ($\rho_1 \oplus \rho_0$) and gripper open width ($\rho_0$), we have $\rho_a = \rho_0^5 \oplus \rho_1^3 \oplus \rho_2$.

## D  Network Architecture Detail

In the image version, we implement the equivariant observation encoder with an equivariant ResNet [58] for the agent view image, a standard ResNet [59] for the eye-in-hand image, and an equivariant MLP for the robot states. We implement the equivariant layers in the group $C_8$. Figure 7 shows the detailed network architecture of our Equivariant Diffusion Policy in the simulation experiments. The network is defined under group $C_8$. First, in the encoding phase, the agent view image is processed with an equivariant ResNet-18, whose output is a $128 \times 8$-dimensional regular representation vector of group. A non-equivariant ResNet-18 with a spatial maxpool at the end processes the eye-in-hand image and outputs a $128$-dimensional representation vector that uses the trivial invariance representation. Those two vectors are concatenated with the gripper position (represented

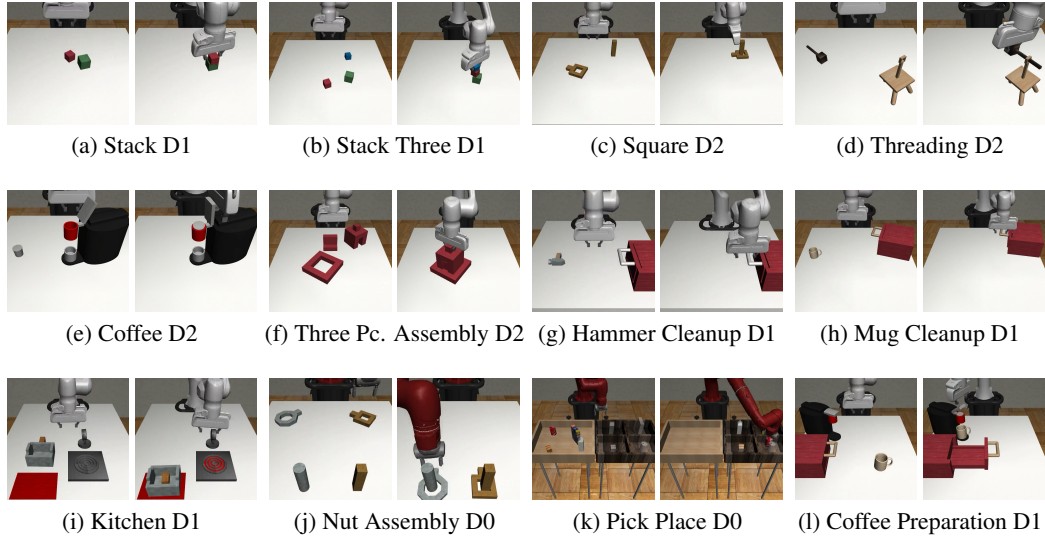

(a) Stack D1     (b) Stack Three D1     (c) Square D2     (d) Threading D2

(e) Coffee D2     (f) Three Pc. Assembly D2    (g) Hammer Cleanup D1     (h) Mug Cleanup D1

(i) Kitchen D1     (j) Nut Assembly D0     (k) Pick Place D0     (l) Coffee Preparation D1

Figure 8: The experimental environments from MimicGen [11]. The left image in each sub figure shows an initial state of the environment; the right image shows the goal state.

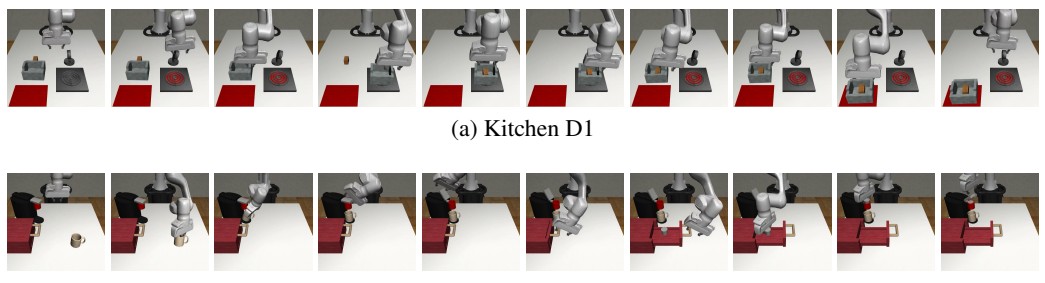

(a) Kitchen D1

(b) Coffee Preparation D1

Figure 9: Illustration of an episode of Kitchen D1 task and Coffee Preparation D1 task.

using $\rho_1 \oplus \rho_0$), gripper orientation (in the format of 6D rotation, represented using $\rho_1^3$), and the gripper finger position (represented using $\rho_0^2$). The concatenated mixed-representation vector is sent to an equivariant linear layer, whose output is a $128 \times 8$-dimensional regular representation observation embedding. The noisy action is also encoded using an equivariant linear layer, whose output is a $64 \times 8$-dimensional regular representation action embedding. Second, in the denoising phase, we process each part of the observation embedding and the action embedding that corresponds to the same group element with a 1D Temporal UNet with hidden dimensions of $[512, 1024, 2048]$ to get a 64-dimensional vector. Doing so for each pair, we will recover a $64 \times 8$-dimensional regular representation noise embedding. In the end, an equivariant linear layer will decode the noise.

In the voxel version, the agent view image is replaced with a voxel grid, and we replace the equivariant ResNet with an 8-layer 3D equivariant convolutional encoder. The 1D Temporal UNet has a hidden dimensions of $[256, 512, 1024]$. The other part of the network stays the same. In the real-world, we remove the eye-in-hand image and only use the voxel grid as vision input (the gripper state vector stays the same).

## E   Simulation Environments

Figure 8 shows the initial and goal states of each tasks. Figure 9 shows an example trajectory for finishing the Kitchen and Coffee Preparation tasks. The RGB observation is an agent-view image and an eye-in-hand image with a size of $3 \times 84 \times 84$. The voxel grid observation has a size of

| Task | Max Steps | Max Out of Plane Rot in Demo |
|---|---|---|
| Stack D1 | 400 | 11.2 |
| Stack Three D1 | 400 | 13.2 |
| Square D2 | 400 | 14.7 |
| Threading D2 | 400 | 13.4 |
| Coffee D2 | 400 | 14.1 |
| Three Piece Assembly D2 | 500 | 16.2 |
| Hammer Cleanup D1 | 500 | 16.4 |
| Mug Cleanup D1 | 500 | 13.0 |
| Kitchen D1 | 800 | 16.2 |
| Nut Assembly D0 | 500 | 15.5 |
| Pick Place D0 | 1000 | 18.0 |
| Coffee Preparation D1 | 800 | 59.0 |

Table 4: The maximum number of time steps and the maximum out of plane rotation (in degrees) in the demo for each simulation environments. The maximum out of plane rotation in the demo is the maximum angular difference between the $SO(3)$ rotation and the $SO(2)$ rotation (i.e., only rotating around the $z$ axis) over all demonstration steps, averaged over 1000 demonstration episodes.

| Method | Obs | Obs Step | Action Pred. Step | Action Exec. Step |
|---|---|---|---|---|
| EquiDiff (Vo) | Voxel Grid, Eye-In-Hand Image, Gripper State | 1 | 16 | 8 |
| EquiDiff (Im) | Agent View Image, Eye-In-Hand Image, Gripper State | 2 | 16 | 8 |
| DiffPo-C | Agent View Image, Eye-In-Hand Image, Gripper State | 2 | 16 | 8 |
| DiffPo-T | Agent View Image, Eye-In-Hand Image, Gripper State | 2 | 10 | 8 |
| DP3 | Point Cloud, Gripper State | 2 | 16 | 8 |
| ACT | Agent View Image, Eye-In-Hand Image, Gripper State | 1 | 10 | 10 |
| BC-RNN | Agent View Image, Eye-In-Hand Image, Gripper State | 1 | 1 | 1 |

Table 5: The observation format, observation step, action prediction step, and action execution step for all methods. The gripper state is a vector including a 3 dimensional position vector, a rotation vector in the format of 6D rotation representation or 4D quaternion, and a 2 dimensional finger position.

$4 \times 64 \times 64 \times 64$ where the first channel is binary occupancy and the remaining three channels are RGB. The point cloud observation has a size of $1024 \times 6$ (i.e, xyzrgb). The point cloud only contains points above the table, as suggested in [20]. All tasks have a full 6 DoF SE(3) action space. Table 4 shows the maximum number of time steps (following [11]) and the maximum out of plane rotation in the demo, calculated by taking the maximum angular difference between the $SO(3)$ rotation and the $SO(2)$ rotation (i.e., only rotating around the $z$ axis) across the entire demonstration episode. Results averaged for 1000 demonstrations.

## F   Training Detail

In the simulation experiments, we follow the hyper-parameters of the prior work [1] for the image version of our method, where the only change is that we increase the batch size to 128 for faster training. Specifically, the observation contains two steps of history observation, and the output of the denoising process is a sequence of 16 action steps. We use all 16 steps for training but only execute eight steps in evaluation. We train our models with the AdamW [60] optimizer (with a learning rate of $10^{-4}$ and weight decay of $10^{-6}$) and Exponential Moving Average (EMA). We use a cosine learning rate scheduler with 500 warm-up steps. We use DDPM [14] with 100 denoising steps for both training and evaluation. For each different number of demos (100, 200, 1000), we maintain roughly the same number of training steps by the training for $50000/n$ epochs where $n$ is the number of demos. Evaluations are conducted every $1000/n$ epochs (50 evaluations in total). In the voxel version, we use only one step of history observation, and keep the other hyper-parameters the same.

The hyper-parameters for the diffusion policy and BC RNN baselines exactly follow [1]. We follow the original work [20] for the hyper-parameters of DP3, except that we use the same action sequence length (16 for training and 8 for evaluation) as [1] and our method. For the ACT baseline, we follow the hyper-parameters provided in the prior work [51], except that we use a chunk size of 10, KL

| Method | Ctrl | Stack D1 | | | Stack Three D1 | | | Square D2 | | | Threading D2 | | |
|---|---|---|---|---|---|---|---|---|---|---|---|---|---|
| | | 100 | 200 | 1000 | 100 | 200 | 1000 | 100 | 200 | 1000 | 100 | 200 | 1000 |
| EquiDiff (Vo) | Abs | 98.7±0.7 | 100.0±0.0 | 100.0±0.0 | 74.7±4.4 | 91.3±0.7 | 90.7±1.3 | 38.7±1.3 | 48.0±3.1 | 63.3±1.3 | 38.7±0.7 | 52.7±2.9 | 54.7±2.9 |
| EquiDiff (Im) | | 93.3±0.7 | 100.0±0.0 | 100.0±0.0 | 54.7±5.2 | 77.3±1.8 | 96.0±1.2 | 25.3±8.7 | 41.3±9.8 | 60.0±4.2 | 22.0±1.2 | 40.0±1.2 | 59.3±1.8 |
| DiffPo-C [1] | | 76.0±4.0 | 97.3±0.7 | 100.0±0.0 | 38.0±0.0 | 72.0±2.0 | 94.0±1.2 | 8.0±1.2 | 19.3±5.3 | 46.0±7.2 | 17.3±1.8 | 35.3±1.3 | 58.7±0.7 |
| DiffPo-T [1] | | 51.3±1.8 | 82.7±0.7 | 98.7±0.7 | 16.7±0.7 | 41.3±2.9 | 84.0±1.2 | 4.7±1.8 | 11.3±2.4 | 44.7±4.7 | 10.7±0.7 | 18.0±1.2 | 40.7±0.7 |
| DP3 [20] | | 69.3±3.7 | 86.7±4.7 | 99.3±0.7 | 7.3±0.7 | 22.7±3.7 | 65.3±1.8 | 6.7±0.7 | 6.0±0.0 | 19.3±3.3 | 12.0±3.1 | 23.3±3.3 | 40.0±2.0 |
| ACT [51] | | 34.7±0.7 | 72.7±7.7 | 96.0±1.2 | 6.0±2.3 | 36.7±2.7 | 78.0±1.2 | 6.0±0.0 | 18.0±1.2 | 49.3±4.7 | 10.0±1.2 | 20.7±2.9 | 35.3±2.4 |
| EquiDiff (Vo) | Rel | 94.7±1.3 | 100.0±0.0 | 100.0±0.0 | 59.3±0.7 | 76.0±0.0 | 82.7±0.7 | 24.7±1.8 | 34.7±5.2 | 52.0±2.3 | 33.3±1.8 | 38.7±2.9 | 46.0±1.2 |
| EquiDiff (Im) | | 74.7±5.8 | 96.0±0.0 | 100.0±0.0 | 25.3±3.3 | 62.7±3.5 | 92.0±1.2 | 11.3±1.3 | 20.7±4.1 | 48.0±4.0 | 11.3±1.3 | 22.0±1.2 | 49.3±2.4 |
| DiffPo-C [1] | | 80.7±2.4 | 93.3±0.7 | 99.3±0.7 | 26.0±4.0 | 52.0±2.0 | 86.0±1.2 | 6.0±1.2 | 13.3±1.3 | 36.7±4.8 | 13.3±1.8 | 26.0±3.1 | 40.0±2.3 |
| BC RNN [2] | | 59.3±7.0 | 94.7±1.3 | 100.0±0.0 | 12.0±2.5 | 48.0±5.3 | 92.0±2.3 | 8.0±1.2 | 20.7±2.7 | 58.7±3.5 | 7.3±0.7 | 13.3±2.4 | 46.7±0.7 |

| Method | Ctrl | Coffee D2 | | | Three Pc. Assembly D2 | | | Hammer Cleanup D1 | | | Mug Cleanup D1 | | |
|---|---|---|---|---|---|---|---|---|---|---|---|---|---|
| | | 100 | 200 | 1000 | 100 | 200 | 1000 | 100 | 200 | 1000 | 100 | 200 | 1000 |
| EquiDiff (Vo) | Abs | 64.7±0.7 | 73.3±1.8 | 76.0±0.0 | 37.3±2.7 | 58.0±5.0 | 71.3±3.3 | 70.0±2.0 | 66.0±2.3 | 72.7±0.7 | 52.7±1.3 | 64.7±2.4 | 68.0±1.2 |
| EquiDiff (Im) | | 60.0±2.0 | 79.3±1.3 | 76.0±2.0 | 15.3±1.8 | 39.3±1.8 | 69.3±3.5 | 65.3±0.7 | 63.3±4.4 | 76.7±0.7 | 49.3±0.7 | 64.0±1.2 | 66.7±0.7 |
| DiffPo-C [1] | | 44.0±1.2 | 66.0±2.3 | 78.7±0.7 | 4.0±0.0 | 6.0±1.2 | 30.0±1.2 | 52.0±1.2 | 58.7±1.3 | 73.3±2.4 | 42.7±0.7 | 58.7±1.3 | 65.3±2.4 |
| DiffPo-T [1] | | 47.3±0.7 | 60.7±1.8 | 74.7±2.7 | 0.7±0.7 | 4.0±0.0 | 42.7±1.3 | 48.0±1.2 | 60.0±1.2 | 76.0±1.2 | 30.0±1.2 | 42.7±2.9 | 63.3±0.7 |
| DP3 [20] | | 34.0±4.0 | 45.3±4.1 | 68.7±2.4 | 0.0±0.0 | 0.7±0.7 | 3.3±0.7 | 54.0±3.1 | 70.7±4.1 | 86.7±0.7 | 21.3±2.7 | 32.7±1.8 | 52.7±4.4 |
| ACT [51] | | 19.3±2.4 | 33.3±2.4 | 64.0±2.3 | 0.0±0.0 | 3.3±0.7 | 24.0±3.1 | 38.0±4.2 | 54.0±1.2 | 70.7±1.3 | 23.3±0.7 | 31.3±1.3 | 56.0±2.0 |
| EquiDiff (Vo) | Rel | 55.3±0.7 | 59.3±0.7 | 64.0±0.0 | 4.7±0.7 | 5.3±0.7 | 54.7±3.5 | 64.0±1.2 | 62.0±1.2 | 67.3±1.3 | 39.3±0.7 | 43.3±1.8 | 62.0±1.2 |
| EquiDiff (Im) | | 40.7±0.7 | 58.7±1.8 | 66.0±1.2 | 1.3±0.7 | 4.7±0.7 | 59.3±4.8 | 48.7±2.7 | 52.0±3.5 | 68.7±2.4 | 29.3±2.9 | 36.0±1.2 | 65.3±2.4 |
| DiffPo-C [1] | | 42.7±1.8 | 50.7±1.8 | 66.7±2.9 | 2.0±1.2 | 2.0±0.0 | 20.0±1.2 | 43.3±1.8 | 54.0±1.2 | 65.3±1.8 | 25.3±0.7 | 39.3±1.8 | 54.7±0.7 |
| BC RNN [2] | | 37.0±1.0 | 52.0±2.0 | 76.0±2.3 | 0.0±0.0 | 5.3±0.7 | 27.0±1.0 | 32.0±0.0 | 42.7±0.7 | 72.0±2.3 | 19.3±0.7 | 39.0±1.0 | 66.7±0.7 |

| Method | Ctrl | Kitchen D1 | | | Nut Assembly D0 | | | Pick Place D0 | | | Coffee Preparation D1 | | |
|---|---|---|---|---|---|---|---|---|---|---|---|---|---|
| | | 100 | 200 | 1000 | 100 | 200 | 1000 | 100 | 200 | 1000 | 100 | 200 | 1000 |
| EquiDiff (Vo) | Abs | 85.3±0.7 | 89.3±0.7 | 88.0±2.3 | 67.3±0.9 | 77.0±0.0 | 83.3±0.7 | 57.7±1.8 | 68.5±0.6 | 82.2±0.8 | 80.0±1.2 | 83.3±1.8 | 85.3±1.8 |
| EquiDiff (Im) | | 67.3±0.7 | 76.7±3.3 | 81.3±0.7 | 74.0±1.2 | 85.0±1.5 | 93.7±0.9 | 41.7±3.2 | 74.2±3.2 | 92.0±1.2 | 76.7±0.7 | 82.7±0.7 | 85.3±0.7 |
| DiffPo-C [1] | | 66.7±2.4 | 84.7±0.7 | 86.7±1.8 | 54.7±2.3 | 68.0±2.6 | 83.0±1.5 | 35.3±2.2 | 65.0±2.8 | 82.7±0.6 | 65.3±0.7 | 62.0±4.2 | 58.0±3.1 |
| DiffPo-T [1] | | 54.0±2.3 | 75.3±0.7 | 81.3±2.4 | 30.7±5.0 | 32.3±5.2 | 45.7±5.9 | 14.7±1.5 | 36.5±1.3 | 50.0±6.0 | 38.0±2.0 | 51.3±1.8 | 76.0±6.0 |
| DP3 [20] | | 44.7±1.8 | 71.3±2.4 | 91.3±2.4 | 15.7±1.3 | 23.7±3.4 | 57.7±1.9 | 11.7±0.9 | 15.0±1.7 | 34.0±0.0 | 10.0±2.3 | 22.0±5.3 | 63.3±4.1 |
| ACT [51] | | 37.3±3.5 | 60.7±3.5 | 87.3±3.5 | 42.3±2.9 | 63.7±3.5 | 84.3±0.9 | 7.2±0.9 | 17.2±1.1 | 50.0±2.9 | 32.0±2.0 | 46.0±3.1 | 64.7±2.4 |
| EquiDiff (Vo) | Rel | 69.3±1.8 | 82.7±1.3 | 89.3±1.8 | 53.0±1.0 | 65.0±2.0 | 72.0±2.0 | 40.3±1.6 | 58.2±0.9 | 78.8±0.8 | 48.0±1.2 | 70.7±2.9 | 73.3±1.8 |
| EquiDiff (Im) | | 60.7±1.3 | 72.0±3.1 | 82.7±2.7 | 44.3±1.2 | 65.3±1.5 | 87.3±0.9 | 29.3±3.1 | 54.7±1.5 | 91.3±1.2 | 48.7±1.3 | 59.3±2.4 | 79.3±0.7 |
| DiffPo-C [1] | | 42.0±2.3 | 64.0±5.0 | 81.3±1.3 | 41.7±2.7 | 62.0±1.5 | 75.3±1.2 | 34.7±1.1 | 58.7±1.0 | 82.2±2.5 | 42.0±3.1 | 52.7±3.3 | 51.3±1.8 |
| BC RNN [2] | | 31.3±2.9 | 46.7±6.7 | 80.7±1.3 | 35.3±0.7 | 58.0±2.1 | 85.0±1.2 | 21.2±0.7 | 41.0±9.0 | 77.3±2.5 | 14.0±1.2 | 32.0±1.2 | 60.7±4.1 |

Table 6: The performance of our Equivariant Diffusion Policy compared with the baselines in simulation. We experiment with 100, 200, and 1000 demos in each environment and report the maximum task success rate among 50 evaluations throughout training. Results averaged over three seeds. ± indicates standard error.

weight of 10, batch size of 64 with learning rate of $5 \times 10^{-5}$, and no temporal aggregation, following the tuning tips provided by the authors. See Table 5 for the observation format, observation step, action prediction step, and action execution step for all methods.

In the real-world experiments, we use a batch size of 64, one step of observation, and disable the EMA. We use DDIM [56] with 100 denoising steps for training and 16 denoising steps for evaluation. The other hyper-parameters stay the same as in simulation.

## G   Simulation Experiment Result with Standard Error

Table 6 shows the same result in Table 1 with the standard error.

## H   Ablation Study

We perform an ablation study regarding the equivariant structure and the voxel input in our method. We consider the following four candidates: 1) Ours: our Equivariant Diffusion Policy with voxel input; 2) Ours no Voxel: our Equivariant Diffusion Policy with RGB input; 3) Ours no Equi.: the baseline Diffusion Policy with voxel input; 4) Ours no Voxel no Equi.: the baseline Diffusion Policy with RGB input, same as [1]. Table 7 shows the result and Table 8 shows the average over all 12 environments. Though both the equivariant structure and the voxel input contribute to the performance improvement of our method, the equivariant structure plays a more important rule, as removing it

| Ablation | Method | Ctrl | Obs | Stack D1 | | | Stack Three D1 | | | Square D2 | | | Threading D2 | | |
|---|---|---|---|---|---|---|---|---|---|---|---|---|---|---|---|
| | | | | 100 | 200 | 1000 | 100 | 200 | 1000 | 100 | 200 | 1000 | 100 | 200 | 1000 |
| - | EquiDiff (Vo) | | Voxel | 99 | 100 | 100 | 75 | 91 | 91 | 39 | 48 | 63 | 39 | 53 | 55 |
| No Voxel | EquiDiff (Im) | Abs | RGB | 93 | 100 | 100 | 55 | 77 | 96 | 25 | 41 | 60 | 22 | 40 | 59 |
| No Equi. | DiffPo-C (Vo) | | Voxel | 87 | 99 | 100 | 33 | 79 | 94 | 10 | 24 | 60 | 19 | 43 | 54 |
| No Voxel No Equi. | DiffPo-C [1] | | RGB | 76 | 97 | 100 | 38 | 72 | 94 | 8 | 19 | 46 | 17 | 35 | 59 |

| Ablation | Method | Ctrl | Obs | Coffee D2 | | | Three Pc. Asse. D2 | | | Hammer Cleanup D1 | | | Mug Cleanup D1 | | |
|---|---|---|---|---|---|---|---|---|---|---|---|---|---|---|---|
| | | | | 100 | 200 | 1000 | 100 | 200 | 1000 | 100 | 200 | 1000 | 100 | 200 | 1000 |
| - | EquiDiff (Vo) | | Voxel | 65 | 73 | 76 | 37 | 58 | 71 | 70 | 66 | 73 | 53 | 65 | 68 |
| No Voxel | EquiDiff (Im) | Abs | RGB | 60 | 79 | 76 | 15 | 39 | 69 | 65 | 63 | 77 | 49 | 64 | 67 |
| No Equi. | DiffPo-C (Vo) | | Voxel | 50 | 72 | 75 | 2 | 5 | 50 | 54 | 64 | 76 | 47 | 58 | 66 |
| No Voxel No Equi. | DiffPo-C [1] | | RGB | 44 | 66 | 79 | 4 | 6 | 30 | 52 | 59 | 73 | 43 | 59 | 65 |

| Ablation | Method | Ctrl | Obs | Kitchen D1 | | | Nut Assembly D0 | | | Pick Place D0 | | | Coffee Prep. D1 | | |
|---|---|---|---|---|---|---|---|---|---|---|---|---|---|---|---|
| | | | | 100 | 200 | 1000 | 100 | 200 | 1000 | 100 | 200 | 1000 | 100 | 200 | 1000 |
| - | EquiDiff (Vo) | | Voxel | 85 | 89 | 88 | 67 | 77 | 83 | 58 | 69 | 82 | 80 | 83 | 85 |
| No Voxel | EquiDiff (Im) | Abs | RGB | 67 | 77 | 81 | 74 | 85 | 94 | 42 | 74 | 92 | 77 | 83 | 85 |
| No Equi. | DiffPo-C (Vo) | | Voxel | 82 | 87 | 87 | 66 | 77 | 84 | 41 | 67 | 84 | 65 | 75 | 77 |
| No Voxel No Equi. | DiffPo-C [1] | | RGB | 67 | 85 | 87 | 55 | 68 | 83 | 35 | 65 | 83 | 65 | 62 | 58 |

Table 7: The ablation study that ablates the voxel input and the equivariant structure in our method. We experiment with 100, 200, and 1000 demos in each environment and report the maximum task success rate among 50 evaluations throughout training. Results averaged over three seeds.

| Ablation | Method | Ctrl | Average over 12 Environments | | |
|---|---|---|---|---|---|
| | | | 100 | 200 | 1000 |
| - | EquiDiff (Vo) | | 63.9 | 72.6 | 77.9 |
| No Voxel | EquiDiff (Im) | Abs | 53.7 (-10.3) | 68.5 (-4.1) | 79.7 (+1.8) |
| No Equi. | DiffPo-C (Vo) | | 46.3 (-17.6) | 62.5 (-10.1) | 75.6 (-2.3) |
| No Voxel No Equi. | DiffPo-C [1] | | 42.0 (-21.9) | 57.8 (-14.8) | 71.4 (-6.5) |

Table 8: The average performance over 12 tasks of the ablation study. Number in parenthesis shows the performance difference after removing different components in our Equivariant Diffusion Policy with voxel input.

(No Equi.) lead to a more significant performance drop compared with removing the voxel input (No Voxel). Note that by using the voxel input, Diffpo-C (Vo) is marginally better than the original Diffusion Policy (DiffPo-C), thus we use Diffpo-C (Vo) as the baseline in our robot experiment in Section 5.3.

## I  Implementing Equivariance via Data Augmentation

In this section, we evaluate implementing equivariance through data augmentation instead of using equivariant networks. Specifically, we applied random rotation data augmentation based on our analysis in Sections 4.1 and 4.2 to a standard, unconstrained CNN. We then compared this CNN + Aug baseline against using equivariant neural networks (Equi. Net) and not implementing equivariance at all (CNN).

As is shown in Table 9, CNN + Aug can achieve good performance, occasionally even outperforming equivariant networks in simpler tasks like Stack and Stack Three. However, it performs poorly in more challenging tasks. When averaged across 12 environments, CNN + Aug performs better than CNN but still underperforms compared to Equi. Net by a significant margin.

## J  $\mathrm{SE}(2)$ Action Space Variation

In this section, we evaluate a variation of our Equivariant Diffusion Policy in an $\mathrm{SE}(2)$ (with $z$ translation) action space to demonstrate the necessity of leveraging an $\mathrm{SE}(3)$ action space. Specifically, the $\mathrm{SE}(2)$ agent only learns the top-down rotation and the out-of-plane rotations will be constantly set to 0. As is shown in Table 10, the $\mathrm{SE}(2)$ variation achieves a similar performance as the $\mathrm{SE}(3)$ version in Stack Three, as the demonstration data in this task has the least amount of out-of-plane

| Method | Obs | Stack D1 | Stack Three D1 | Square D2 | Threading D2 |
|---|---|---|---|---|---|
| Equi. Net (Vo) | Voxel | 98.7±0.7 | 74.7±4.4 | 38.7±1.3 | 38.7±0.7 |
| CNN + Aug (Vo) | Voxel | 99.3±0.7 | 84.0±1.2 | 36.0±1.2 | 30.7±1.8 |
| CNN (Vo) | Voxel | 86.7±1.3 | 33.3±1.8 | 10.0±2.0 | 19.3±2.4 |
| Equi. Net (Im) | RGB | 93.3±0.7 | 54.7±5.2 | 25.3±8.7 | 22.0±1.2 |
| CNN + Aug (Im) | RGB | 98.7±0.7 | 68.0±1.2 | 26.7±1.8 | 22.0±1.2 |
| CNN (Im) | RGB | 76.0±4.0 | 38.0±0.0 | 8.0±1.2 | 17.3±1.8 |

| Method | Obs | Coffee D2 | Three Pc. Asse. D2 | Hammer Cleanup D1 | Mug Cleanup D1 |
|---|---|---|---|---|---|
| Equi. Net (Vo) | Voxel | 64.7±0.7 | 37.3±2.7 | 70.0±2.0 | 52.7±1.3 |
| CNN + Aug (Vo) | Voxel | 56.7±2.9 | 7.3±0.7 | 70.7±1.8 | 52.0±1.2 |
| CNN (Vo) | Voxel | 50.0±3.1 | 2.0±0.0 | 54.0±3.1 | 46.7±0.7 |
| Equi. Net (Im) | RGB | 60.0±2.0 | 15.3±1.8 | 65.3±0.7 | 49.3±0.7 |
| CNN + Aug (Im) | RGB | 58.0±1.2 | 5.3±0.7 | 61.3±2.9 | 50.0±1.2 |
| CNN (Im) | RGB | 44.0±1.2 | 4.0±0.0 | 52.0±1.2 | 42.7±0.7 |

| Method | Obs | Kitchen D1 | Nut Assembly D0 | Pick Place D0 | Coffee Prep. D1 |
|---|---|---|---|---|---|
| Equi. Net (Vo) | Voxel | 85.3±0.7 | 67.3±0.9 | 57.7±1.8 | 80.0±1.2 |
| CNN + Aug (Vo) | Voxel | 62.0±2.0 | 51.7±1.5 | 39.5±2.8 | 48.7±2.4 |
| CNN (Vo) | Voxel | 82.0±2.3 | 66.0±1.7 | 40.8±1.9 | 65.3±0.7 |
| Equi. Net (Im) | RGB | 67.3±0.7 | 74.0±1.2 | 41.7±3.2 | 76.7±0.7 |
| CNN + Aug (Im) | RGB | 47.3±2.9 | 53.7±0.7 | 27.7±0.8 | 34.7±2.9 |
| CNN (Im) | RGB | 66.7±2.4 | 54.7±2.3 | 35.3±2.2 | 65.3±0.7 |

| Method | Obs | Average over 12 Environments |
|---|---|---|
| Equi. Net (Vo) | Voxel | 63.9 |
| CNN + Aug (Vo) | Voxel | 53.3 |
| CNN (Vo) | Voxel | 46.3 |
| Equi. Net (Im) | RGB | 53.7 |
| CNN + Aug (Im) | RGB | 46.2 |
| CNN (Im) | RGB | 42.0 |

Table 9: Comparing implementing equivariance via equivariant network or data augmentation. We experiment with 100 demos in each environment and report the maximum task success rate among 50 evaluations throughout training. Results averaged over three seeds.

| | Stack Three D1 | Threading D2 | Coffee Preparation D1 |
|---|---|---|---|
| EquiDiff (Im), SE(3) Action | 77.3 | 40.0 | 85.3 |
| EquiDiff (Im), SE(2) Action | 75.3 | 12.7 | 0.0 |

Table 10: Performance of Equivariant Diffusion Policy in $SE(2)$ action space compared with $SE(3)$ action space. 200 demos are used in this experiment.

rotation (as shown in Table 4). On the other hand, the $SE(2)$ variation significantly underperforms in Threading, since the ability of wiggling the out-of-plane rotation helps the agent to precisely insert the tool. In the end, the $SE(2)$ agent cannot solve Coffee Preparation at all, because the task requires a significant amount of out-of-plane rotation (as shown in Figure 9b).

# K Robomimic Experiment

In this section, we compare our Equivariant Diffusion Policy with the original Diffusion Policy across four Robomimic tasks (Figure 10). Both methods are trained with 100 Proficient-Human (PH) demonstrations or 100 Multi-Human (MH) demonstrations. Other hyperparameters mirror those used in our MimicGen experiment.

Table 11 shows the result. Our method achieves similar or slightly better performance compared to the baseline Diffusion Policy. The improvements in Robomimic tasks are smaller than in Mimic-Gen tasks. This is because the Robomimic tasks can be classified as Low-Equivariance Tasks (as illustrated in Figure 5, bottom), with minimal randomness in the initial distribution (except for Lift), making the symmetry in our method less advantageous.

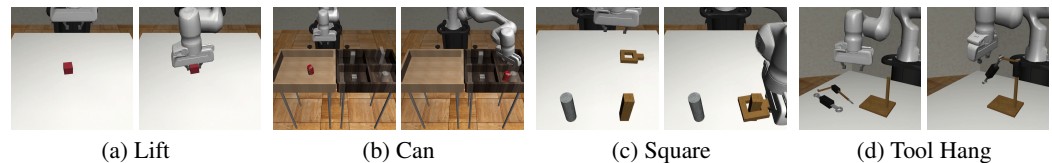

|  | (a) Lift | (b) Can | (c) Square | (d) Tool Hang |
|---|---|---|---|---|

Figure 10: The experimental environments from Robomimic. The left image in each subfigure shows the initial state of the environment; the right image shows the goal state.

|  | Lift | | Can | | Square | | tool hang | Average |
|---|---|---|---|---|---|---|---|---|
|  | 100 PH | 100 MH | 100 PH | 100 MH | 100 PH | 100 MH | 100 PH |  |
| EquiDiff | 100.0±0.0 | 100.0±0.0 | 99.3±0.7 | 96.7±0.7 | 84.0±1.2 | 76.7±1.3 | 76.0±0.0 | 90.4±2.3 |
| DiffPo | 100.0±0.0 | 100.0±0.0 | 100.0±0.0 | 95.3±0.7 | 85.3±0.7 | 70.7±0.7 | 64.0±5.8 | 87.9±3.2 |

Table 11: The performance of our Equivariant Diffusion Policy compared with the Diffusion Policy baseline in Robomimic. We experiment with 100 Proficient-Human (PH) or Multi-Human (MH) demos in each environment and report the maximum task success rate among 50 evaluations throughout training. Results averaged over three seeds. ± indicates standard error.

## L    Real-Robot Environment Details

Figure 11 shows our real-world experimental platform containing a Franka Emika [53] and three Intel Realsense[55] D455 cameras. Compared with simulation, we use a pair of fin-ray [54] gripper fingers instead of the original Franka fingers. Figure 6 shows the five tasks in this experiment. In Oven Opening, the oven is randomly initialized at one of the four borders of the workspace. In Banana in Bowl, the initial poses of the banana and the bowl are both randomly sampled. In Trash Sweeping, the robot needs to use a tool brush to sweep two pieces of crumpled paper out of its workspace. The initial poses of the objects are randomly sampled. In Letter Alignment, the robot needs to align the letters to form "AI". The letter A is randomly initialized at one of the four corners of the workspace, and the pose of the I is randomly sampled. In Hammer to Drawer, the robot needs to open a drawer, pick up a hammer, place it inside the drawer, and close the drawer. The drawer is initialized at one of the four borders of the workspace, and the hammer is randomly initialized at the opposite side of the drawer. Lastly, we also evaluate a Bagel Baking task with an extremely long time horizon, where the robot needs to open the oven, pull out the tray inside the oven, pick up the bagel, place it inside the tray, close the tray, and close the oven. In this task, the oven is randomly initialized at one of the three borders of the workspace (where we eliminate the side that is furthest from the robot to avoid joint limits of the robot), and the bagel is randomly initialized along the opposite side of the oven. The observation is a voxel grid with a resolution of $64 \times 64 \times 64$ and the gripper pose and open width. The voxel grid covers the $(0.4m)^3$ workspace. During training, we apply a random crop augmentation to crop the voxel grid to $58 \times 58 \times 58$. In Banana in Bowl and Trash Sweeping, we train the model with an additional random rotation augmentation. The baseline is trained with the same data augmentation as our method.

## M    Generalization Experiment

In this experiment, we evaluate the generalizability of our Equivariant Diffusion Policy to unseen object poses. We conduct this evaluation in the Bagel Baking experiment in the real world, where the oven is initialized in three different poses during training (Fig 12a). At test time, we rotate the oven to eight different, unseen poses (Fig 12b). We found that the learned policy can zero-shot generalize to these unseen rotations, with the exception of the scenario where the oven is rotated to the bottom-right corner. In this case, the policy is constrained by the robot's joint limits. Specifically, the policy was able to open the oven and pull out the tray, but when picking up the bagel, although the policy could generate good gripper poses, the actions were infeasible for the robot due to joint limits. This generalization demonstrates the power of the equivariant structure in our policy.

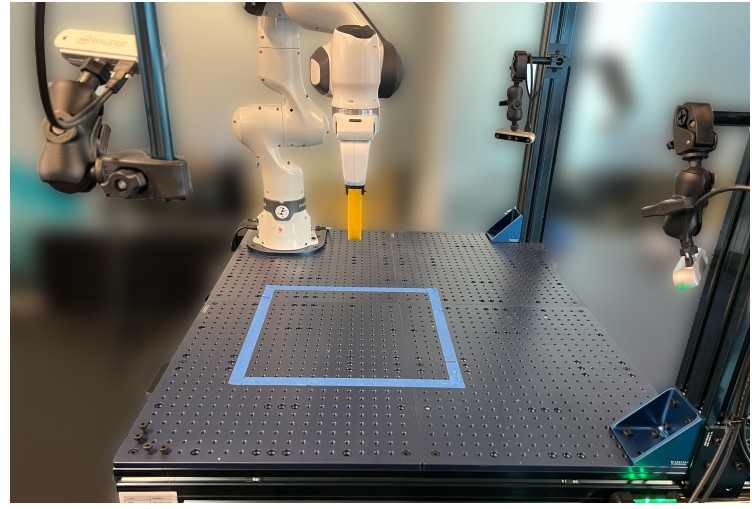

Figure 11: Our real-robot platform contains a Franka Emika robot arm equipped with a pair of fin-ray fingers, and three Intel Realsense D455 cameras.

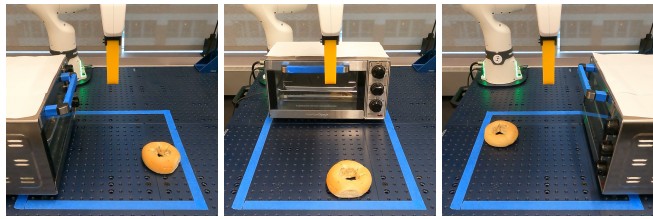

(a) Oven Poses in Training Set

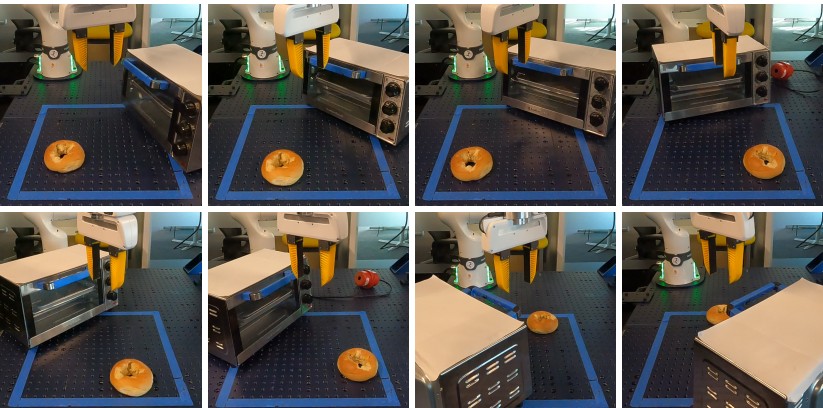

(b) Oven Poses in The Generalization Experiment

Figure 12: (a) The initial oven poses in the training set. (b) The oven poses in the generalization experiment. Those poses are unseen during training. In both training and testing, the pose of the bagel is random.

