# OpenReview forum: "Equivariant Diffusion Policy"
_robot-learning.org/CoRL/2024/Conference — CoRL 2024_

### Official Review · Reviewer_TQso · 2024-07-14

**Originality:** 4
**Technical Quality:** 5
**Clarity Of Presentation:** 4
**Potential Impact:** 3
**Recommendation:** 4
**Confidence:** 4

**Review:**

## Paper strengths and contributions
**Motivation**
- The motivation for leveraging SO(2)-equivariant to improve Diffusion Policy is convincing.

**Technical contribution**
- This work analyzes the equivariance in the diffusion process for policy learning in detail and presents an implementation that could learn an equivariant Diffusion Policy.

**Clarity**
- The overall writing is clear. The authors utilize figures well to illustrate the ideas.

**Experimental results**
- The experiments in both simulation (12 manipulation tasks from MimicGen) and a real-world system (Franka Emika robot arm) show that the proposed framework outperforms existing methods, including Diffusion Policy, 3D Diffusion Policy, Action Chunking Transformer, and BC-RNN, in terms of the performance of learned policies as well as data efficiency, i.e., the amount of demonstration required.

## Paper weaknesses and questions

**Limited to robot arm manipulation**
- The analysis and evaluation mainly focus on robot arm manipulation. It would be interesting to see if such equivariance, while additional considerations might be needed, could also improve the performance of diffusion policies in navigation or locomotion tasks.

**Error bars are missing**
- Tables 1-2 and Figure 5(b) fail to report error bars, e.g., the standard deviation of the success rate, making it difficult to understand the performance. Are the results statistically significant?

**Related work**
- It would be great if the authors could discuss some recent works that explore using diffusion models for imitation learning, including the following papers, which can make the contributions of this work clearer. I am aware that some of the works were made public around or even after the CoRL submission deadline, so it was impossible for the authors to include them. Therefore, not discussing these works does not at all affect my evaluation of this work's novelty and contributions.
    - "Constrained-Context Conditional Diffusion Models for Imitation Learning" (TMLR) focuses on resolving the spurious correlation issue of diffusion policies.
    - "Diffusion model-augmented behavioral cloning" (ICML 2024) learns a diffusion model to model expert state-action pairs and then provides gradients to train a policy.
    - "DiffAIL: Diffusion Adversarial Imitation Learning" (AAAI 2023) and "Diffusion-Reward Adversarial Imitation Learning" (2024) use conditional diffusion models as GAIL discriminators.
    - "EquiBot: SIM(3)-Equivariant Diffusion Policy for Generalizable and Data Efficient Learning" (2024)

**Quality Of The Limitations Section:**

2

**Questions For Rebuttal:**

See above

**Robotics Focus:**

4

**Summary Of Paper:**

This paper aims to leverage domain symmetries in robot arm manipulation tasks to improve the performance and sample efficiency of Diffusion Policies. Specifically, the proposed framework uses the SO(2)-equivariant of planar rotations, i.e., rotation around the z-axis of the world), employing an equivariant observation encoder, an equivariant action encoder, and an equivariant action decoder. The experiments in both simulation (12 manipulation tasks from MimicGen) and a real-world system (Franka Emika robot arm) show that the proposed framework outperforms existing methods, including Diffusion Policy, 3D Diffusion Policy, Action Chunking Transformer, and BC-RNN, in terms of the performance of learned policies as well as data efficiency, i.e., the amount of demonstration required. This work is well-motivated, tackles an important problem, and presents a reasonable framework with sufficient evaluation. I believe this work will interest a wide range of audiences at CoRL and therefore recommend accepting this work.

**Summary Of Recommendation:**

This work is well-motivated, tackles an important problem, and presents a reasonable framework with sufficient evaluation. I believe this work will interest a wide range of audiences at CoRL and therefore recommend accepting this work.

---

### Official Review · Reviewer_1QLT · 2024-07-21

**Originality:** 3
**Technical Quality:** 3
**Clarity Of Presentation:** 3
**Potential Impact:** 3
**Recommendation:** 3
**Confidence:** 4

**Review:**

Improving data efficiency is crucial for robotics policy learning, and using equivariance is a promising direction. This paper does a commendable job in terms of method description and experiments. However, to enhance the research, several limitations and potential improvements should be considered:
1. While SO(2) rotation about the z-axis on a 3D representation (i.e., voxel) is logical, its application to 2D images is less intuitive, especially if the image is not top-down. In such cases, the rotated image and rotated actions may lose consistency. More detailed illustrations would be beneficial to clarify this aspect.
2. Besides equivariant neural networks, data augmentation offers another straightforward approach to incorporate equivariance by applying the same transformation to both observation (i.e., voxel) and action. It would be valuable to present results using the original architecture trained on an augmented dataset for comparison.
3. Given that only 8 discrete rotations are considered, it's worth exploring whether the policy could generalize to continuous rotation for initial poses. For instance, how would the policy perform if the oven were rotated 30 degrees in the bagel task? The provided oven video only demonstrates rotations of 0, 90, and 180 degrees.
4. It would be helpful to explain the rationale behind using tasks from MimicGen instead of Robomimic, which was utilized in the original diffusion policy paper. This choice might impact the comparability of results between the two studies.

**Quality Of The Limitations Section:**

2

**Questions For Rebuttal:**

Listed in the "Review" section.

**Robotics Focus:**

4

**Summary Of Paper:**

This paper introduces equivariant diffusion policy, which improves the performance of the original diffusion policy, especially in the low-data regime. To this end, the major contribution is action equivariance in SO(2) space. Finally, extensive experimental results in both simulation and the real world are included to show the performance improvement.

**Summary Of Recommendation:**

It has good quality and could be potentially improved.

---

### Official Review · Reviewer_T42w · 2024-07-21
**Clever usage of equivariance to improve success rate and sample efficiency when learning from demonstrations.**

**Originality:** 4
**Technical Quality:** 5
**Clarity Of Presentation:** 5
**Potential Impact:** 4
**Recommendation:** 4
**Confidence:** 2

**Review:**

The paper is very well written and easy to follow. The authors introduced the necessary concepts to understand equivariance in the context of their work. Moreover, they did a great job of clearly explaining their method, especially how equivariance is achieved in the action space.

The motivation is clear and depicted with a simple example to make it understandable for the reader.

The major theoretical contribution is the adaptation of the action representation into the SO(2) format. I am not an expert in equivariance in SO(2), so maybe other reviewers could comment on if the math behind is sound.

The simulation and real-world experiments are very well constructed. The baselines are carefully chosen to choose the benefit of using equivariance in the network structure. A very important aspect was to show that equivariance is crucial to achieve comparable success rates in low-data regimes, which are common in robot learning from demonstrations. It is interesting to notice that the performance difference gets smaller for higher amounts of demonstrations, which is expected since they probably cover more parts of the observation and action spaces. I believe Figure 5 very well summarises the paper's results. It is clear that tasks where an equivariance policy is crucial largely benefit from introducing this network structure in the learned diffusion policy.

**Quality Of The Limitations Section:**

3

**Questions For Rebuttal:**

One of the benefits of diffusion models is that they model highly multimodal distributions quite well.
Compared to the diffusion policy, do you notice any drop in multimodality due to the equivariant networks?

Since your observation encoder is equivariant, do you still need to do data augmentation for the voxel and image-based observations?
If yes, did you also use data augmentation when training the (unstructured) diffusion policy?

In the introduction, you state: “…we theoretically demonstrate the use of SO(2)-equivariance in the context of 6-DoF control for robotic manipulation, which prior methods [9, 10] leveraged in a less expressive SE(2) action space…” I’d expect the SE(2) space to be “more expressive” than SO2 since it includes translations. Could you clarify why you say SE(2) is less expressive?

**Robotics Focus:**

4

**Summary Of Paper:**

The authors propose Equivariant Diffusion Policy, an imitation learning policy based on diffusion policy that is SE(3) equivariant, i.e., if the observation inputs undergo an homogeneous transformation H, the output actions also expected to undergo the same transformation.  Importantly, the authors consider visual inputs and not state based ones, which shows the higher relevance of this work for real world applications.

**Summary Of Recommendation:**

This paper is a great contribution to the field of structured policies for imitation learning. The theory is sound and clearly explained. The experiments, both in simulation and real world, showcase the benefit of using equivariant representations over the baselines. Most importantly, the authors show that using equivariant networks improves greatly on sample efficiency, which is a bottleneck for learning from demonstrations.

---

### Author Rebuttal · Authors · 2024-08-07

We thank the AC and all reviewers for their careful and helpful reviews and comments. In the rebuttal, we have conducted several new experiments to address the reviewers’ feedback, and we believe these new results will improve our paper. Here is a summary of the rebuttal:
1. Added a new experiment comparing data augmentation against equivariant networks (T42w, 1QLT).
2. Added a Robomimic experiment (pAe1, 1QLT).
3. Added a generalization experiment in the real-world, showing that the learned policy can generalize to novel initial poses (pAe1, 1QLT).
4. Added the error bars in the simulation experiment (TQso).

Please refer to the new experiments in the rebuttal file, as well as the video of the generalization experiment. We will include these new experiments in the final version of our paper.

---

### Decision · Program_Chairs · 2024-09-04

**Decision:**

Accept

**Comment:**

# Strength
- All reviewers are mostly aligned on the technical strength of the paper

# Weakness
- No major weakness highlighted by the reviewers

# Recommendation
Reviewers have made excellent recommendations to convert a good to a great paper.
- Why MimicGen instead of Robomimic?
- Demonstration on locomotion tasks on uneven terrains. Additionally validates its effectiveness on nonflat surfaces.
- Extension to continuous rotations.

Authors are highly encouraged to work with the reviewers to accommodate these suggestions to further strengthen the submission.

# Rebuttal
All reviewers are aligned on their recommendations post rebuttal.